# IMiDs induce FAM83F degradation via an interaction with CK1α to attenuate Wnt signalling

Karen Dunbar, Thomas J Macartney, Gopal P Sapkota

Immunomodulatory imide drugs (IMiDs) bind CRBN, a substrate receptor of the Cul4A E3 ligase complex, enabling the recruitment of neo-substrates, such as CK1α, and their degradation via the ubiquitinproteasome system. Here, we report FAM83F as such a neo-substrate. The eight FAM83 proteins (A-H) interact with and regulate the subcellular distribution of CK1α. We demonstrate that IMiD-induced FAM83F degradation requires its association with CK1α. However, no other FAM83 protein is degraded by IMiDs. We have recently identified FAM83F as a mediator of the canonical Wnt signalling pathway. The IMiD-induced degradation of FAM83F attenuated Wnt signalling in colorectal cancer cells and removed CK1α from the plasma membrane, mirroring the phenotypes observed with genetic ablation of FAM83F. Intriguingly, the expression of FAM83G, which also binds to CK1α, appears to attenuate the IMiD-induced degradation of CK1α, suggesting a protective role for FAM83G on CK1α. Our findings reveal that the efficiency and extent of target protein degradation by IMiDs depends on the nature of inherent multiprotein complex in which the target protein is part of.

## Introduction

Thalidomide, the first immunomodulatory imide drug (IMiD), initially came to prominence as a treatment for morning sickness in the 1950s but was quickly abandoned after it became apparent that consumption of thalidomide in the first trimester of pregnancy caused foetal abnormalities, predominately manifesting as limb deformities (Vargesson, 2015). Despite these severe teratogenic effects, the mechanism of action remained elusive for several decades until it was found that IMiDs hijack the ubiquitin–proteasomal system to facilitate protein degradation of non-native substrates, which have been termed "neo-substrates" (Kronke et al, 2014). IMiDs act as molecular glues by binding to both neo-substrates and a hydrophobic binding pocket of cereblon (CRBN), which is a substrate receptor of the Cul4A–E3 ligase complex. This brings the neo-substrates into close proximity to the Cul4A–ROC1–DDB1–CRBN E3 ligase complex (known as Cul4A^CRBN),

thereby facilitating their ubiquitylation and subsequent proteasomal degradation (Kronke et al, 2014). Recently, two distinct derivative analogues of thalidomide, lenalidomide (Rajkumar et al, 2005) and pomalidomide (Miguel et al, 2013), have been repurposed for the effective treatment of multiple myeloma. Their efficacy has been attributed to the induced degradation of the zinc-finger transcription factors IKZF1 and IKZF3 which have key roles in B- and T-cell biology (Kronke et al, 2014).

Whereas the majority of identified IMiD neo-substrates appear to be zinc-finger transcription factors (Kronke et al, 2014; An et al, 2017; Sievers et al, 2018), lenalidomide has also been shown to induce the degradation of the serine/threonine kinase CK1α (Kronke et al, 2015). Casein kinase 1 isoforms (α, α-like, δ, ε, γ1, γ2, and γ3) are a family of serine/threonine protein kinases which control many cellular processes, including Wnt signalling, circadian rhythms, calcium signalling, cell division, and responses to DNA damage (Cheong & Virshup, 2011; Cruciat, 2014; Jiang et al, 2018; Philpott et al, 2020). Lenalidomide binds to a β-hairpin loop in the kinase N-lobe of CK1α, bringing it into proximity of the Cul4A^CRBN complex to facilitate its ubiquitylation and subsequent proteasomal degradation (Petzold et al, 2016). The degradation of CK1α is thought to cause the efficacy of lenalidomide in the treatment of myelodysplastic syndromes (MDS) (List et al, 2006). MDS are a group of blood cancers, of which a subtype are caused by deletion of chromosome 5q (del(5q)) (List et al, 2006). In such cancers, deletion of a region of chromosome 5q results in CK1α haploinsufficiency through loss of the *CSNK1A1* gene (Kronke et al, 2015), thereby sensitizing cells against further degradation of CK1α by lenalidomide.

Historically, CK1 isoforms were thought to be monomeric, unregulated, and constitutively active, but there is now accumulating evidence that a family of previously uncharacterised proteins, the FAM83 proteins, act as anchors for several of the CK1 isoforms (α, α-like, δ, and ε) and can alter their subcellular localisation in response to specific stimuli (Bozatzi et al, 2018; Fulcher et al, 2018). The FAM83 family is composed of 8 members, termed FAM83A-H, which share a conserved N-terminal domain of unknown function 1669 (DUF1669), which mediates the interaction with different CK1 isoforms (Fulcher et al, 2018). Each member binds to different CK1 isoforms with varying specificity and affinity (Fulcher et al, 2018). All FAM83 proteins interact with CK1α, whereas FAM83A, B, E, and H also

Medical Research Council Protein Phosphorylation and Ubiquitylation Unit, School of Life Sciences, University of Dundee, Sir James Black Centre, Dundee, UK

Correspondence: g.sapkota@dundee.ac.uk

interact with CK1δ and ε (Fulcher et al, 2018). Whereas the FAM83 family remains largely uncharacterised, roles for specific FAM83–CK1α complexes have been established in mitosis (Fulcher et al, 2019; Fulcher & Sapkota, 2020) and canonical Wnt signalling (Bozatzi et al, 2018; Wu et al, 2019; Dunbar et al, 2020). Given the reports of IMiD-induced degradation of CK1α, we sought to establish the effect of IMiDs on the stability of FAM83 proteins and different FAM83–CK1α complexes.

# Results

### IMiDs selectively degrade FAM83F protein

Lenalidomide, which is used as a therapeutic agent in patients with del(5q) MDS, causes CK1α degradation (Kronke et al, 2015). In MV4.11 cells, which are derived from B-myelomonocytic leukaemia, the thalidomide derivatives lenalidomide, pomalidomide, and iberdomide led to robust degradation of IKZF1, but only lenalidomide, and to a lesser extent pomalidomide, led to partial degradation of CK1α (Fig 1A). The efficacy of lenalidomide-induced CK1α degradation in other cell lines, including THP-1 monocytes, HCT116 colorectal cancer, A549 lung adenocarcinoma, DLD-1 colorectal cancer, PC-3 prostate cancer, and HaCaT keratinocyte cell lines were more variable with the most substantial CK1α degradation observed in HCT116, DLD-1, and HaCaT cells (Fig 1B). As the FAM83 proteins exist in complexes with CK1α (Fulcher et al, 2018), we sought to test the effect of the IMiDs thalidomide, lenalidomide, and pomalidomide on FAM83 protein levels in THP-1, HCT116, A549, DLD-1, PC-3, HaCaT, U2OS, HEK-293, ARPE-19, SH-SY5Y, G-361, and SK-MEL-13 cell lines. Both lenalidomide and pomalidomide but not thalidomide induced a robust reduction in FAM83F protein abundance in HCT116, DLD-1, and HaCaT cells, whereas in the other tested cell lines FAM83F protein was not detectable (Figs 1C and D and S1A and B). None of the IMiDs led to any detectable reduction in FAM83B, FAM83D, FAM83G, or FAM83H protein abundance in any cell line tested (Figs 1C and S1A and B). Currently, no reliable antibodies exist for the detection of endogenous FAM83A, FAM83C, and FAM83E proteins, limiting their assessment in this assay. A modest degradation of CK1α was observed with lenalidomide treatment in THP-1, HCT116, A549, HEK-293, SH-SY5Y, and G-361 cells (Figs 1C and D and S1A and B), whereas no consistent change in either CK1δ or CK1ε protein levels was detected following IMiD treatment (Figs 1C and S1A and B). As expected, lenalidomide and pomalidomide caused robust degradation of IKZF1 in THP-1 cells, whereas no IKZF1 was detected in non-hematopoietic cells (Fig 1C). ZFP91, a pomalidomide specific neo-substrate (An et al, 2017), was degraded upon pomalidomide treatment in all cell lines, whereas its expression was undetectable in THP-1 cells (Figs 1C and S1A and B).

Further characterisation of FAM83F degradation in HCT116 and DLD-1 cells revealed a time and dose dependence on lenalidomide and pomalidomide treatment, with optimal degradation occurring after 24-h treatments with 10 $\mu$M IMiD (Figs 1E and F and S2A and B). Novel heterobifunctional compounds that recruit target proteins to CRBN through an IMiD moiety, such as dTAG-13, lead to target protein degradation through Cul4A$^{CRBN}$ (Nabet et al, 2018). We found

that dTAG-13 was incapable of degrading CK1α, FAM83F, and ZFP91 in HCT116 and DLD-1 cells, confirming the utility of certain IMiD moieties for development of bivalent protein degraders (Fig S2C) (Nabet et al, 2018).

### FAM83F and CK1α protein abundance is reduced at the plasma membrane upon IMiD treatment

Endogenous FAM83F is predominantly located at the plasma membrane as observed by immunofluorescence of HCT116 cells in which a GFP tag was knocked in homozygously at the N-terminus of the *FAM83F* gene ($^{GFP/GFP}$FAM83F cells) (Figs 2A, S3A, and S4A) (Dunbar et al, 2020). Upon treatment of HCT116 $^{GFP/GFP}$FAM83F cells with pomalidomide, the GFP signal was lost from the plasma membrane (Fig 2A). Under basal conditions, CK1α, which can interact with all eight FAM83 proteins, is distributed throughout the cell, and so we were unable to determine the effect of pomalidomide on membranous CK1α by immunofluorescence (Fig 2A). However, when we analysed subcellular fractions of DLD-1 wild-type cells, we observed that FAM83F was predominately present in the membrane fraction, with a small proportion also observed in the nuclear fraction, whereas CK1α is present in cytoplasmic, nuclear, and membrane fractions (Fig 2B). When DLD-1 wild-type cells were treated with pomalidomide, a reduction in the levels of both FAM83F and CK1α protein was observed in the membrane fraction, whereas CK1α protein levels in the cytoplasmic and nuclear fractions did not change relative to untreated DLD-1 wild-type cells (Fig 2B). FAM83F interacts with CK1α and is responsible for delivering it to the plasma membrane (Dunbar et al, 2020). Interestingly, the pomalidomide-induced reduction of CK1α protein from the plasma membrane in DLD-1 wild-type cells was comparable with the stable reduction of membranous CK1α observed in FAM83F-knockout (FAM83F$^{-/-}$) DLD-1 cells, generated using CRISPR/Cas9 genome editing, in the absence of pomalidomide treatment (Figs 2B, S3A, and S4B). In the absence of FAM83F, pomalidomide treatment did not induce any detectable degradation of CK1α from the membrane fraction in DLD-1 FAM83F$^{-/-}$ cells, confirming that the IMiD-induced loss of CK1α protein from the plasma membrane is mediated by FAM83F.

### IMiD-induced degradation of FAM83F requires interaction with CK1α

The robust degradation of FAM83F by lenalidomide and pomalidomide in several cancer cell lines prompted us to explore whether this degradation was mediated through the association of FAM83F with CK1α. Several conserved residues within the DUF1669 domains of FAM83 proteins have been identified as critical mediators of the FAM83-CK1 interaction (Fulcher et al, 2018). For FAM83F, mutation of two phenylalanine residues at positions 284 and 288 to alanine would be predicted to abolish association with CK1α (Dunbar et al, 2020). As expected, FAM83F co-precipitated with CK1α in CK1α immunoprecipitates (IPs) from HCT116 wild-type cells but not from FAM83F-knockout (FAM83F$^{-/-}$) cells generated using CRISPR/Cas9 (Figs 3A, S3A, and S4B). FAM83F also co-precipitated with CK1α in CK1α IPs from HCT116 FAM83F$^{-/-}$ cells that were stably rescued with FAM83F$^{WT}$ but not from HCT116 FAM83F$^{-/-}$ cells rescued with FAM83F$^{F284A/F288A}$ mutant (Fig 3A).

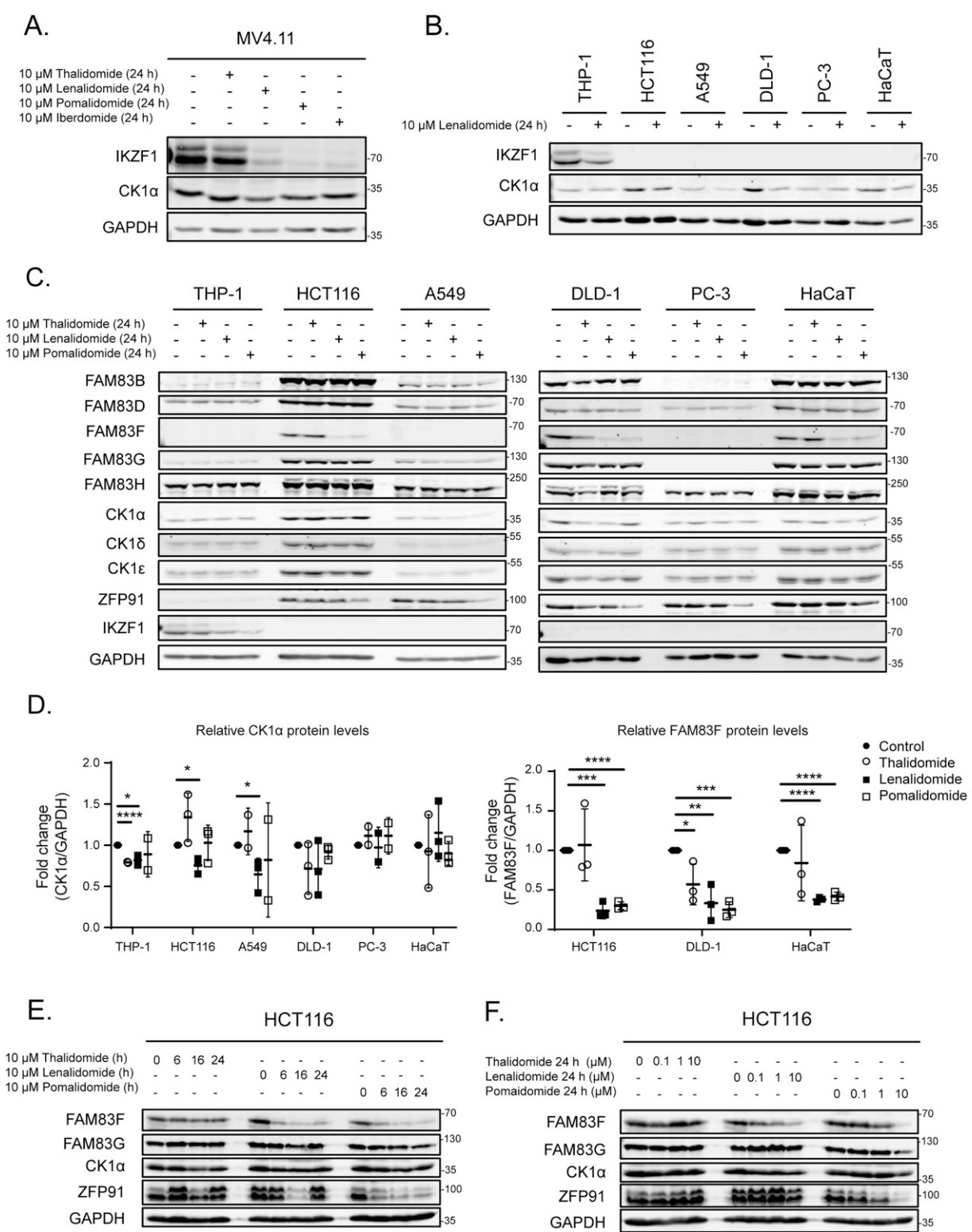

**Figure 1. IMiDs degrade FAM83F but no other FAM83 proteins.**
**(A)** MV4.11 cell extracts, treated with or without the IMiDs (10 $\mu$M for 24 h), were resolved by SDS–PAGE and subjected to Western blotting with the indicated antibodies.
**(B)** THP-1, HCT116, A549, DLD-1, PC-3, and HaCaT cell extracts treated with or without 10 $\mu$M lenalidomide for 24 h were resolved by SDS–PAGE and subjected to Western blotting with the indicated antibodies. **(C)** THP-1, HCT116, A549, DLD-1, PC-3, and HaCaT cell extracts treated with or without IMiDs (10 $\mu$M for 24 h) were resolved by SDS–PAGE and subjected to Western blotting with the indicated antibodies. **(D)** Densitometry of FAM83F and CK1$\alpha$ protein abundance upon treatment with the indicated IMiDs (10 $\mu$M for 24 h). FAM83F and CK1$\alpha$ protein abundances were normalised to GAPDH protein abundance and represented as fold change compared to untreated cells. Data

The lenalidomide and pomalidomide-induced degradation of FAM83F was only evident in HCT116 wild-type and HCT116 FAM83F$^{-/-}$ cells rescued with FAM83F$^{WT}$, but not in HCT116 FAM83F$^{-/-}$ cells or in HCT116 FAM83F$^{-/-}$ cells rescued with FAM83F$^{F284A/F288A}$ mutant (Fig 3B and C). Overall, these observations suggest that the interaction of FAM83F with CK1α is required for the IMiD-induced degradation of FAM83F (Fig 3B and C).

## IMiD-induced FAM83F degradation is mediated via the proteasome and is dependent on cereblon

IMiD-induced degradation requires the Cul4A$^{CRBN}$ E3 ligase complex for the ubiquitylation of neo-substrates and subsequent proteasomal degradation. Activation of Cul E3 ligases requires NEDDylation of Cullin subunits. Thus, Cul E3 ligase activity can be blocked by inhibiting the catalytic activity of the NEDD8-activating enzyme with the small molecule inhibitor MLN4924 (Duda et al, 2008). Treatment of cells with MLN4924 prevented lenalidomide and pomalidomide-induced degradation of FAM83F in both DLD-1 and HCT116 cells, indicating the requirement of a Cul E3 ligase for degradation (Fig 4A). Inhibition of the proteasome with bortezomib, which leads to the accumulation of mono- and poly-ubiquitylated proteins in cell extracts, also prevented lenalidomide and pomalidomide-induced FAM83F degradation, indicating that the reduction in FAM83F protein is mediated by the proteasome (Fig 4A). To ascertain whether the IMiD-induced degradation of FAM83F was dependent on CRBN, we knocked out CRBN from DLD-1 cells using CRISPR/Cas9 genome editing (Figs S3A–S5B). Lenalidomide and pomalidomide-induced degradation of FAM83F evident in DLD-1 wild-type cells was completely abolished in DLD-1 CRBN$^{-/-}$ cells (Fig 4B and C). Restoration of DLD-1 CRBN$^{-/-}$ cells with human FLAG-CRBN partially restored the lenalidomide and pomalidomide-induced degradation of FAM83F. However, when DLD-1 CRBN$^{-/-}$ cells were rescued with the FLAG-CRBN$^{V388I}$ mutant, which mimics the mouse variant shown to be inactive for IMiD-induced protein degradation as the IMiD is unable to bind CRBN$^{V388I}$ (Kronke et al, 2015), lenalidomide and pomalidomide did not induce FAM83F degradation (Fig 4B and C). These findings confirm that IMiD-induced FAM83F degradation requires the Cul4A$^{CRBN}$ E3 ligase activity and is mediated by the proteasome.

## FAM83G protects CK1α from IMiD-induced degradation

Lenalidomide-induced degradation of CK1α has been shown to be robust in multiple myeloid cells (Kronke et al, 2015). In agreement, we observed more robust degradation of CK1α upon lenalidomide and other IMiD treatments in MV4.11 cells compared with DLD-1 cells (Fig S6A and B) or a panel of other cancer cell lines (Figs 1 and S1). Interestingly, the levels of most FAM83 proteins in MV4.11 cells were either absent (FAM83B & F), or much lower in abundance (FAM83D, G & H) compared with DLD-1 cells (Fig S6A). We hypothesised that the absence of specific FAM83–CK1α complexes may explain why CK1α

degradation with lenalidomide was more robust in MV4.11 cells compared with non-hematopoietic cell lines. To investigate whether specific FAM83 proteins impact IMiD-induced CK1α degradation, we generated HCT116 FAM83F$^{-/-}$ and FAM83G$^{-/-}$ cells with CRISPR/Cas9 genome editing (Figs S3–S5). Both lenalidomide and pomalidomide treatment caused FAM83F degradation in HCT116 wild-type and FAM83G$^{-/-}$ cells, whereas FAM83G levels remained unchanged upon IMiD treatment in both HCT116 wild-type and FAM83F$^{-/-}$ cells (Fig 5A and B). Lenalidomide reduced CK1α protein abundance in HCT116 wild-type, FAM83F$^{-/-}$, and FAM83G$^{-/-}$ cell lines, but interestingly the reduction in FAM83G$^{-/-}$ cells was slightly but significantly higher than in wild-type cells (Fig 5A and B). In addition, thalidomide and pomalidomide were also able to induce a more robust CK1α degradation in FAM83G$^{-/-}$ cells compared with wild-type and FAM83F$^{-/-}$ cells (Fig 5A and B). Considering this increased degradation is observed with multiple IMiDs in FAM83G$^{-/-}$ cells, the possibility that, in the absence of FAM83G, an increased proportion of CK1α binds to FAM83F and is co-degraded as a complex was tested. Indeed, when we overexpressed wild-type GFP-FAM83G in HCT116 cells, pomalidomide-induced degradation of FAM83F was partially rescued suggesting that less CK1α is bound to FAM83F with CK1α preferentially binding to GFP-FAM83G (Fig 5C–F). Thus, FAM83F is partially protected from degradation. Indeed, overexpression of GFP-FAM83G$^{F296A}$, which has minimal interaction with CK1α (Fig 5C and D) (Fulcher et al, 2018) in HCT116 cells was unable to reduce the pomalidomide-induced degradation of FAM83F (Fig 5E and F). However, it must be noted that these changes in protein abundance are only slight and not statistically significant, which is unsurprising as CK1α exists in multiple protein complexes, including in other FAM83–CK1α complexes. Intriguingly, FAM83G was identified by mass spectrometry as one of the top interactors of overexpressed CRBN upon IMiD treatment (Kronke et al, 2014). This would suggest that multiple CK1α–FAM83 complexes could be recruited to the Cul4A$^{CRBN}$ complex after exposure to IMiDs, but only the CK1α–FAM83F complex is positioned in such a way that ROC1 can ubiquitylate both CK1α and FAM83F.

## BTX161 is an efficient CK1α-degrader which reduces FAM83G protein abundance through FAM83G–CK1α co-stability

The success of lenalidomide and pomalidomide in multiple myeloma treatment has led to the continuing development of new IMiD derivatives. BTX161 has been reported to be the most efficient CK1α degrader to date (Minzel et al, 2018). We compared BTX161 with other IMiD compounds and confirmed the potency of BTX161 in CK1α degradation in MV4.11 and DLD-1 cells (Fig S6A and B). We sought to establish the effect of BTX161 on FAM83 protein stability using HCT116, DLD-1, and MV4.11 cells (Fig 6A and B). We observed BTX161-induced degradation of FAM83F in DLD-1 and HCT116 cells (Figs 6A and B and S6A). FAM83F and CK1α degradation in these cells occurred in a time and dose-dependent manner with optimal degradation

presented as scatter graph illustrating individual data points with an overlay of the mean ± SD. **(E)** HCT116 cell extracts, treated with 10 μM IMiDs for 6, 16, or 24 h, were resolved by SDS–PAGE and subjected to Western blotting with the indicated antibodies. **(F)** HCT116 cell extracts, treated with 0.1, 1 or 10 μM of IMiDs for 24 h, were resolved by SDS–PAGE and subjected to Western blotting with the indicated antibodies. Statistical analysis of data was completed using a Student's unpaired t test and comparing fold change between untreated and IMiD treated samples. Statistically significant P-values are denoted by asterisks (* < 0.05, ** < 0.01, *** < 0.001, **** < 0.0001). Source data are available for this figure.

A.

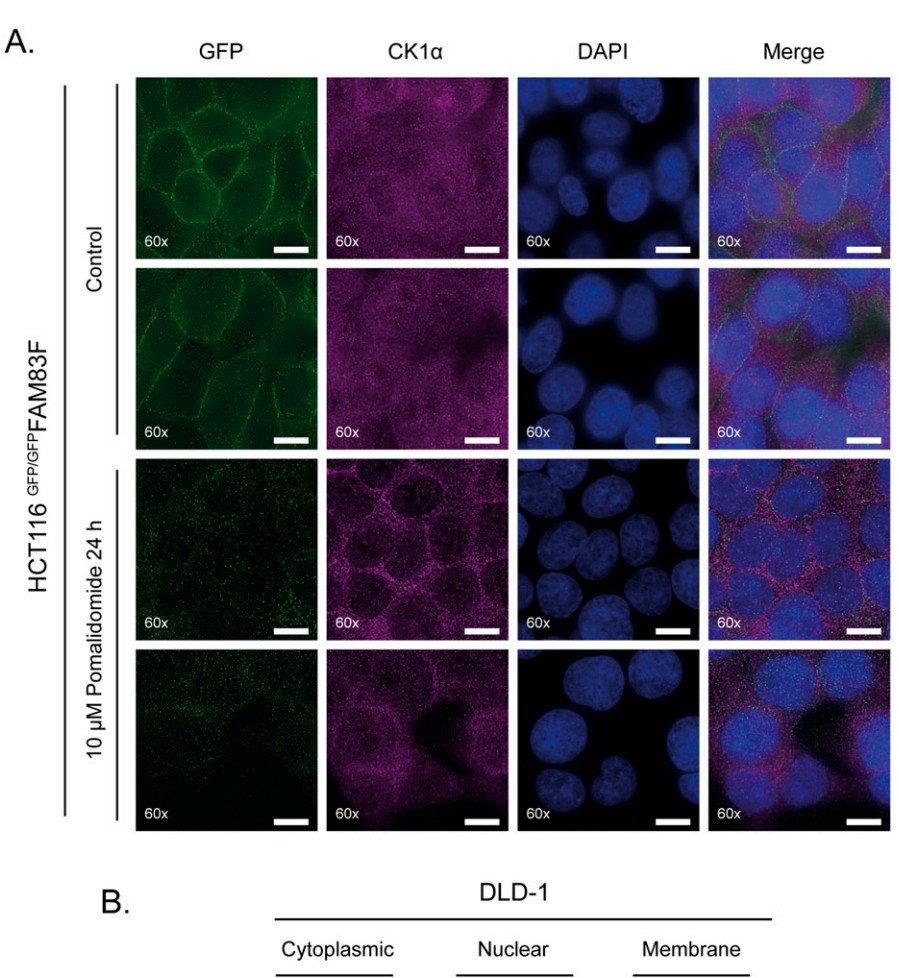

**Figure 2. FAM83F and CK1α protein abundance is reduced at the plasma membrane upon IMiD treatment.**

**(A)** Wide-field immunofluorescence microscopy of HCT116 GFP/GFPFAM83F knock-in cells, treated with or without 10 μM pomalidomide for 24 h, stained with antibodies recognising GFP (far left panels, green), and CK1α (second row of panels from left, magenta) as well as DAPI (third row of panels from left, blue). Overlay of GFP, CK1α, and DAPI images as a merged image is shown on the right. Immunofluorescence images were captured with a 60× objective. Scale bar represents 10 μm. Two representative images for each staining are shown. **(B)** Specific subcellular fractions from cytoplasmic, nuclear, and membrane compartments from DLD-1 wild-type and FAM83F−/− cells treated with or without 10 μM pomalidomide for 24 h, were resolved by SDS–PAGE and subjected to Western blotting with the indicated antibodies. Specificity of cytoplasmic, nuclear and membrane fractions were determined with Western blotting with compartment specific antibodies: α-tubulin (cytoplasmic), Lamin A/C (nuclear), and Na/K-ATPase (membrane).

Source data are available for this figure.

B.

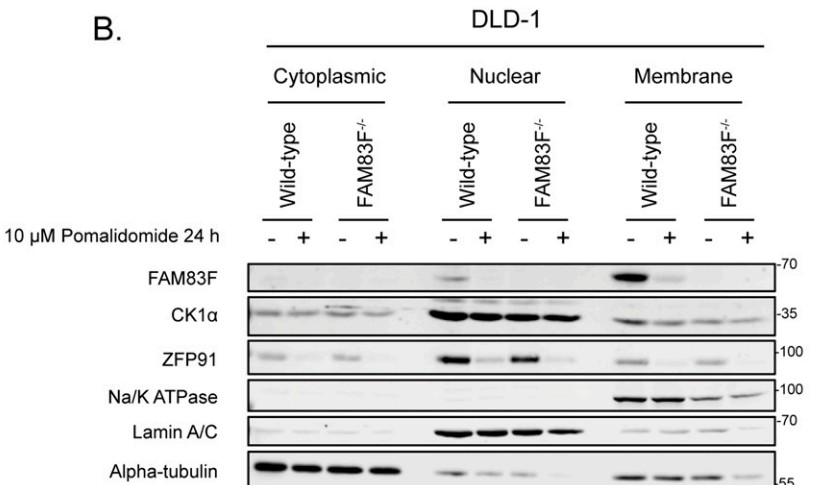

observed after treatment with 10 μM BTX161 for 24 h (Fig 6A and B). CK1α degradation was even more substantial in MV4.11 cells, which do not contain detectable levels of FAM83F protein (Figs 6A and B and S6A and B). We quantified protein levels of CK1α, FAM83F, FAM83H, FAM83G, and FAM83B via Western blot analysis after treatment of HCT116, DLD-1, and MV4.11 cells with 10 μM BTX161 for 24 h (Fig 6C). FAM83F and CK1α abundances were consistently reduced in all cell lines. FAM83H protein abundance was slightly reduced in HCT116 and MV4.11 cells. FAM83B protein abundance was unaffected by BTX161 treatment in all cells. FAM83G protein abundance was slightly reduced in DLD-1 cells but

almost fully depleted in MV4.11 cells, whereas protein levels in HCT116 cells appear unchanged (Fig 6C). We confirmed that this strong FAM83G degradation observed in MV4.11 cells was specific for BTX161 and, to a lesser extent, lenalidomide treatments (Fig S7).

Given that the reductions in FAM83G protein abundance in various cell lines mirror the reduction in CK1α protein, we queried whether the FAM83G–CK1α complex is co-degraded by BTX161, similarly to the FAM83F–CK1α complex, or if this was a result of CK1α-dependent stability of FAM83G. CK1α protein was reduced in HCT116 and DLD-1 cells using CK1α-targeting siRNA (siCK1α) (Fig 6D

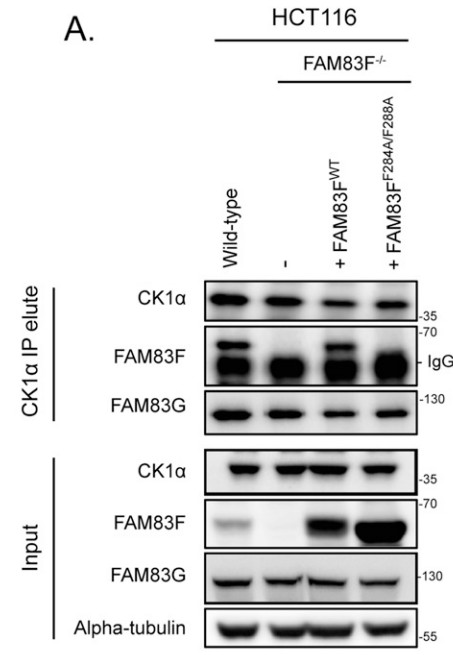

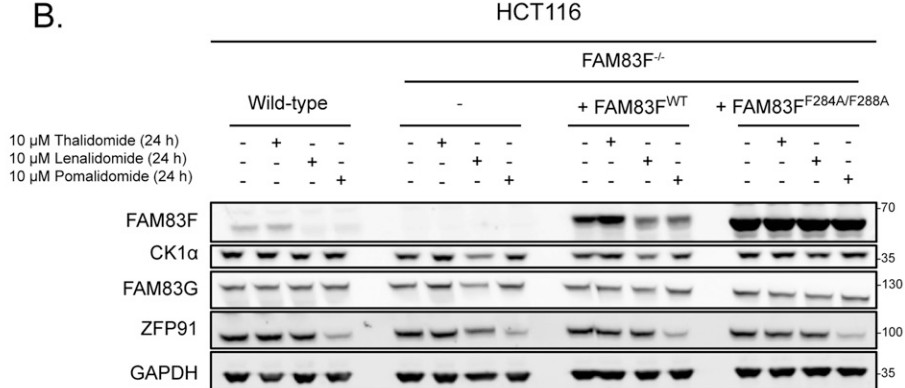

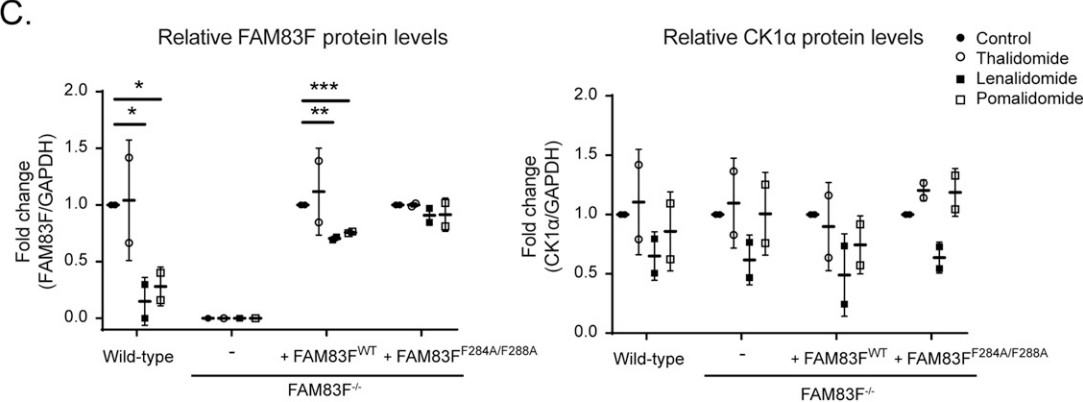

**Figure 3. IMiD-induced degradation of FAM83F requires interaction with CK1α.**
**(A)** HCT116 wild-type, HCT116 FAM83F$^{-/-}$, HCT116 FAM83F$^{-/-}$ rescued with FAM83F$^{WT}$, and HCT116 FAM83F$^{-/-}$ rescued with FAM83F$^{F284A/F288A}$ cell extracts were subjected to immunoprecipitation (IP) with anti-CK1α antibody. Input extracts and CK1α IP elutes were resolved by SDS–PAGE and subjected to Western blotting with the indicated antibodies. **(B)** As in (A), except the cells were treated with IMiDs (10 μM for 24 h) as indicated and then were resolved by SDS–PAGE and subjected to Western blotting with the indicated antibodies. **(C)** Densitometry of FAM83F and CK1α protein abundance upon treatment with IMiDs (10 μM for 24 h) from (B). FAM83F and CK1α protein abundance was normalised to GAPDH protein abundance and represented as fold change compared to untreated cells. Data presented as scatter graph illustrating

and E). This produced an ~75–80% reduction in CK1α protein after 48 h and resulted in a significant reduction in FAM83G protein when compared with non-targeting siRNA controls (Fig 6D and E). Loss of CK1α, caused by either BTX161 treatment or siRNA, resulted in reduced FAM83G protein levels suggesting some form of co-stability between the two proteins. In contrast, FAM83F protein levels were unaffected by siRNA treatment. We therefore propose that only FAM83F–CK1α complex can be degraded by lenalidomide, pomalidomide, and BTX161 but the loss of CK1α by lenalidomide and BTX161 can result in a reduction in FAM83G due to co-stability. The efficiency of CK1α degradation by IMiDs varies between cell lines and may be affected by the abundance of FAM83 proteins, some of which may protect CK1α from IMiD-induced degradation and thus the abundance of specific FAM83 proteins in cells could predict the efficacy of overall CK1α degradation.

### Inducible degradation of FAM83F attenuates Wnt signalling

We have recently established that FAM83F mediates canonical Wnt signalling through association with CK1α (Dunbar et al, 2020). Ablation of FAM83F significantly inhibits Wnt signalling in multiple cell lines. The plasma membrane localisation of FAM83F mediated by its C-terminal farnesylation is also essential for its role in Wnt signalling (Dunbar et al, 2020). Hyperactivated Wnt signalling caused by a truncation of adenomatous polyposis coli (Apc) is a hallmark of colorectal cancer initiation and thus we sought to determine if IMiD-induced FAM83F degradation could dampen down Wnt signalling in DLD-1 cells, which harbour a truncated Apc mutant protein (Yang et al, 2006). We sought to evaluate the ability of pomalidomide to reduce Wnt signalling in Apc mutant cells by assessing the abundance of a Wnt-target gene, *Axin2* (Jho et al, 2002), in DLD-1 wild-type, FAM83F$^{-/-}$, and CRBN$^{-/-}$ cells following pomalidomide treatment (Fig 7A). In addition, IMiD treatment of DLD-1 cells efficiently reduced FAM83F, but not CK1α protein levels (Fig 1C and D). Treatment of DLD-1 cells with 10 μM pomalidomide for 48 h significantly reduced *Axin2* mRNA abundance in wild-type cells. Pomalidomide treatment did not alter *Axin2* mRNA abundance in DLD-1 FAM83F$^{-/-}$ or DLD-1 CRBN$^{-/-}$ cells indicating that the pomalidomide-induced reduction in *Axin2* mRNA requires FAM83F and CRBN. To confirm that treatment with pomalidomide could replicate all phenotypes associated with genetic knockout of FAM83F, membrane fractions from DLD-1 wild-type, FAM83F$^{-/-}$ and CRBN$^{-/-}$ cell lines treated with 10 μM pomalidomide for 24 h were assessed for FAM83F and CK1α protein abundance (Fig 7B). In DLD-1 wild-type cells, membranous FAM83F and CK1α protein abundance was reduced upon pomalidomide treatments, whereas no changes were detected in DLD-1 CRBN$^{-/-}$ cells. CK1α levels in membrane fractions of DLD-1 FAM83F$^{-/-}$ cells, which were already reduced under untreated conditions compared with DLD-1 wild-type cells, were not altered upon pomalidomide treatment.

To determine if pomalidomide could also reduce Wnt signalling in cell lines containing no activating Wnt signalling mutations, we treated HaCaT, HEK-293, and A549 cell lines with pomalidomide and measured *Axin2* mRNA expression and FAM83F protein degradation (Fig 7C and D). Pomalidomide treatment significantly reduced basal *Axin2* mRNA and FAM83F protein expression in HaCaT cells (Fig 7C and D). Pomalidomide treatment increased basal *Axin2* mRNA expression in HEK-293 cells with no effect in A549 cells (Fig 7C and D). Interestingly, HEK-293 and A549 cells do not express detectable levels of FAM83F protein suggesting that the pomalidomide reduction in *Axin2* transcript expression requires FAM83F protein (Fig 7C and D). These results demonstrate that IMiD-induced degradation of FAM83F protein replicates the phenotypes observed with genetic FAM83F-knockout cells in two distinct cell lines and, importantly, IMiD-induced degradation of FAM83F appears to reduce Wnt activity in colorectal cancer cells displaying constitutively active Wnt signalling.

## Discussion

Lenalidomide (Revlimid) was the second highest grossing prescription drug worldwide in 2018 and with clinical trials for additional haematological derived cancers ongoing, the prominence of IMiDs are likely to increase further (Urquhart, 2019). Therefore, the discovery of novel proteins targeted for IMiD-induced degradation is important to predict both unforeseen consequences of IMiD treatments and potential new therapeutic targets. The inducible degradation of CK1α, upon lenalidomide or BTX161 treatment, has been established previously (Kronke et al, 2015; Minzel et al, 2018) (Fig 8A). Here, we report that several IMiD compounds can induce degradation of the FAM83F protein and demonstrate that degradation of FAM83F requires its ability to interact with CK1α, indicating that IMiD-induced recruitment of CK1α to CRBN mediates the co-recruitment of FAM83F (Fig 8B). The specific degradation of FAM83F and the absence of degradation of other FAM83 proteins, after treatment with IMiD compounds, has been corroborated by quantitative mass spectrometry (Donovan et al, 2018). It remains unknown whether all FAM83–CK1α complexes are recruited to the Cul4A$^{CRBN}$ complex upon IMiD treatment but not all complexes are ubiquitylated and degraded. Alternatively, the binding of certain FAM83 proteins to CK1α could restrict the recruitment of CK1α to CRBN. However, given that FAM83G was reported as one of the top binders of CRBN from cells treated with lenalidomide (Kronke et al, 2014), the former hypothesis appears more likely. Intriguingly, we did observe a slight reduction in FAM83G protein abundance after BTX161 treatment. However, given the substantial CK1α degradation observed upon BTX161 treatment and that knockdown of CK1α through siRNA reduces FAM83G abundance, these observations on FAM83G protein stability are likely a result of FAM83G-CK1α co-stability rather than direct degradation of the FAM83G–CK1α complex (Fig 8C).

Increasingly, targeted protein degradation is being used as an investigative tool in cell biology, with hopes to translate into a

individual data points with an overlay of the mean ± SD. Statistical analysis of data was completed using a Student's unpaired *t* test and comparing fold change between untreated and IMiD treated samples. Statistically significant *P*-values are denoted by asterisks (* < 0.05, ** < 0.01, *** < 0.001, **** < 0.0001). Source data are available for this figure.

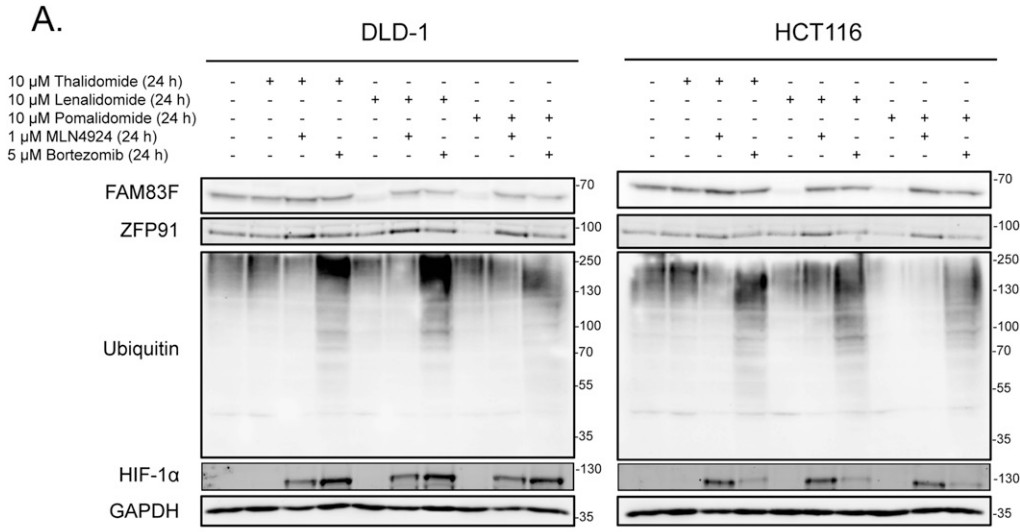

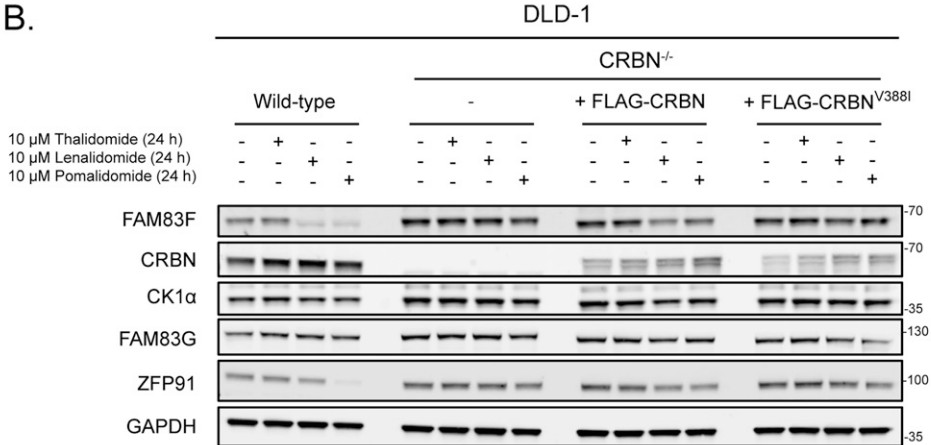

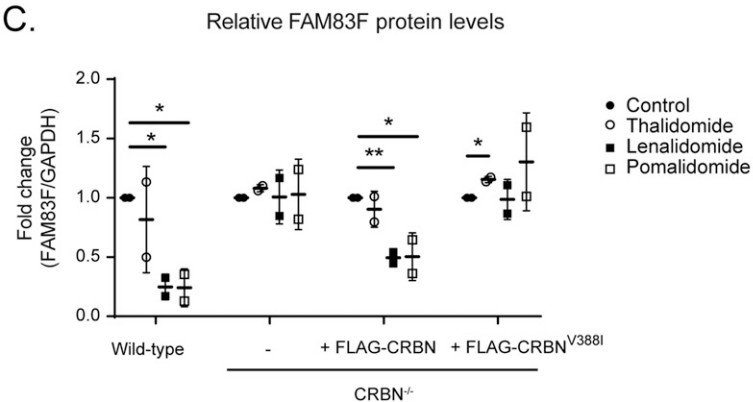

**Figure 4. IMiD-induced FAM83F degradation occurs via the proteasome and is dependent on cereblon.**

**(A)** DLD-1 and HCT116 cell extracts treated with 10 µM IMiDs, 1 µM MLN4924, and 5 µM Bortezomib or a combination thereof as indicated for 24 h were resolved by SDS–PAGE and subjected to Western blotting with the indicated antibodies. The accumulation of HIF-1α and ubiquitylated proteins following MLN4924 and Bortezomib treatments, respectively were used as positive controls for successful compound treatments. **(B)** DLD-1 wild-type, DLD-1 CRBN⁻/⁻, DLD-1 CRBN⁻/⁻ rescued with FLAG-CRBN, and DLD-1 CRBN⁻/⁻ rescued with FLAG-CRBN$^{V388I}$ cell extracts treated with IMiDs (10 µM for 24 h), were resolved by SDS–PAGE and subjected to Western blotting with the indicated antibodies. **(C)** Densitometry of FAM83F protein abundance upon treatment with IMiDs (10 µM for 24 h) from (B). FAM83F protein abundance was normalised to GAPDH protein abundance and represented as fold change compared with untreated cells. Data presented as scatter graph illustrating individual data points with an overlay of the mean ± SD. Statistical analysis of data was completed using a Student's unpaired *t* test and comparing fold change between untreated and IMiD treated samples. Statistically significant *P*-values are denoted by asterisks (* < 0.05, ** < 0.01, *** < 0.001, **** < 0.0001). Source data are available for this figure.

therapeutic option for a variety of diseases (Roth et al, 2019). The efficacy of inducible protein degradation is influenced by a number of factors including protein synthesis rate, binding affinity of protein of interest to E3 ligase complexes, efficiency of ubiquitylation and presence of deubiquitinases (Schapira et al, 2019). Ubiquitylation requires accessible lysine residues in close proximity to the E3 ligase (Hershko & Ciechanover, 1992). Different IMiDs bind neo-substrates at slightly different angles, thus presenting different residues to the

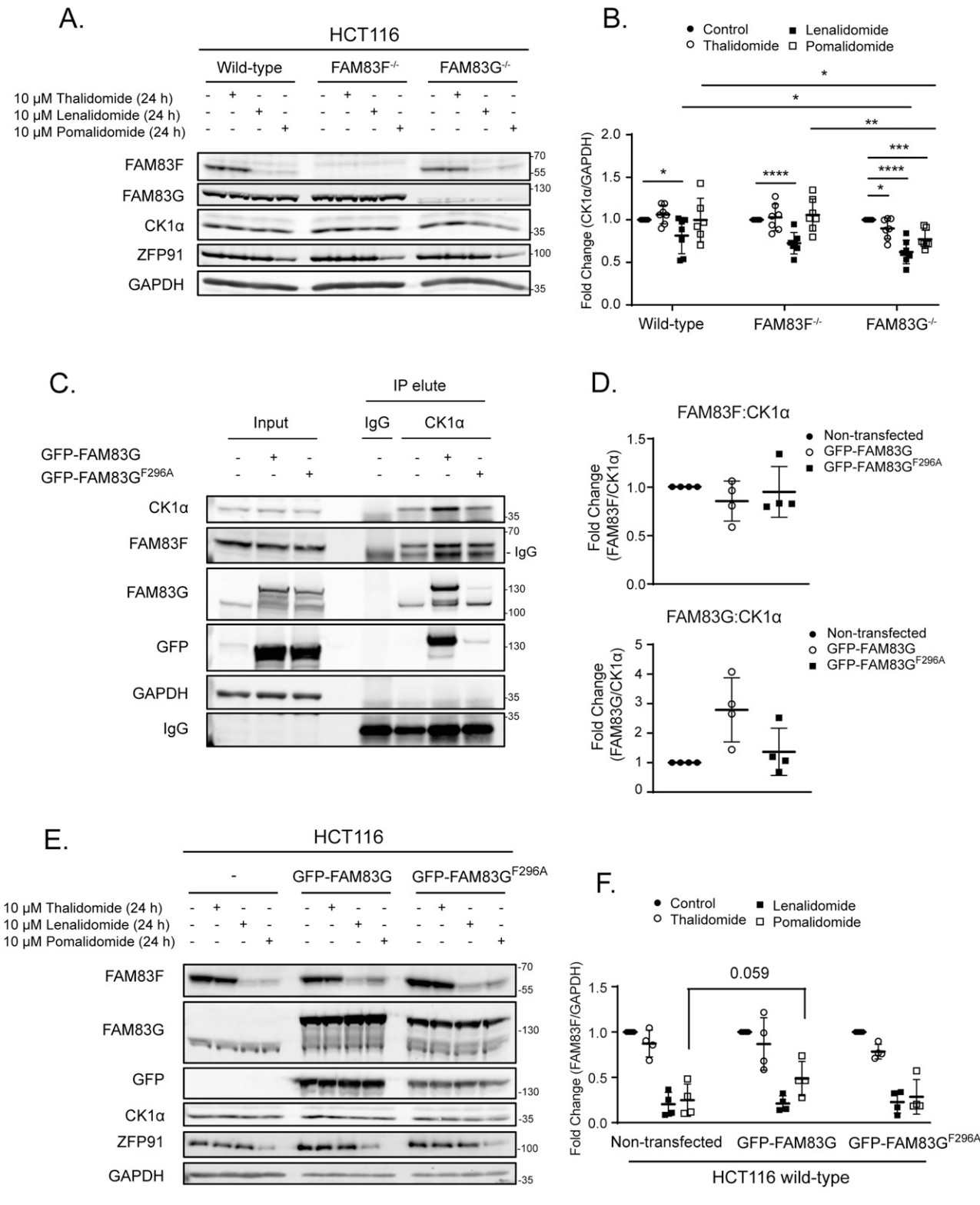

**Figure 5. FAM83G protects CK1α from IMiD-induced degradation.**
**(A)** HCT116 wild-type, HCT116 FAM83F$^{-/-}$ and HCT116 FAM83G$^{-/-}$ cell extracts treated with IMiDs (10 $\mu$M for 24 h) were resolved by SDS–PAGE and subjected to Western blotting with indicated antibodies. **(B)** Densitometry of CK1α protein abundance upon treatment with IMiDs in HCT116 wild-type, HCT116 FAM83F$^{-/-}$, and HCT116 FAM83G$^{-/-}$ cells from (A). CK1α protein abundance was normalised to GAPDH protein abundance and represented as fold change compared to untreated cells. Data presented as scatter graph illustrating individual data points with an overlay of the mean ± SD. **(C)** HCT116 wild-type, HCT116 wild-type transfected with GFP-FAM83G and HCT116 wild-type transfected with GFP-FAM83G$^{F296A}$ cell extracts were subjected to IP with anti-CK1α antibody. Input extracts, and indicated IP elutes were resolved by SDS–PAGE and subjected to Western

ROC1 E3 ligase which may explain the differences in protein degradation noted between various IMiD compounds (Sievers et al, 2018). However, numerous unresolved questions remain regarding targeted proteolysis, including how the formation of different protein complexes affects degradation kinetics and/or occurrence. With our data, we make important observations suggesting that the architecture of CK1α in different FAM83–CK1α complexes most likely determines whether, and to what extent, CK1α and the associated FAM83 protein can be degraded by IMiDs. Specifically, the FAM83F–CK1α complex is robustly degraded by various IMiD compounds, whereas sequestering CK1α in the FAM83G–CK1α complex spares CK1α from IMiD-induced degradation.

Clinically, lenalidomide treatment in del(5q) MDS patients was shown to be effective, and it was suggested that this might be due to haploinsufficiency of CK1α (Kronke et al, 2015). We propose that the efficiency of CK1α degradation is rather influenced by the relative abundance of FAM83 proteins. We observe substantial lenalidomide-induced CK1α degradation in MV4.11 cells, which lack expression of several FAM83 proteins, but not in any other cell line which displays higher abundance of FAM83 proteins. Therefore, we hypothesise that lenalidomide can facilitate the degradation of the non-FAM83 bound pool of CK1α as well as the FAM83F-bound pool and thus the abundance of other FAM83 proteins in cells may be used as predicting biomarkers for levels of IMiD-induced CK1α degradation, which may inform the use of lenalidomide for targeting CK1α. In addition to IMiDs, other target protein degraders such as PROTACs are emerging as key therapeutic modalities in drug research (Paiva & Crews, 2019). Our observations presented in this study clearly illustrate that the nature of the inherent complex in which the target protein exists is an important factor that determines whether the target protein can be degraded. This not only provides challenges in trying to design degraders of specific target proteins that yield complete degradation but also provide opportunities in which specific protein complexes can be targeted for degradation, thereby affecting specific functions of target proteins. In this regard, the efficacy for a protein degrader should perhaps not be judged by how much a target protein is degraded but rather to what extent a change in expected phenotype is achieved. Indeed, for proteins which exist in distinct functional pools, specific degradation of subcomplexes may be sufficient to disrupt target pathways, while leaving other non-targeted subcomplex functions intact.

FAM83F has been implicated in oncogenesis with high FAM83F expression observed in oesophageal squamous cell carcinoma (Mao et al, 2016), lung adenocarcinoma (Fan et al, 2019), glioma (Xu et al, 2018), and thyroid carcinoma (Fuziwara et al, 2019). In contrast, FAM83F has been reported to enhance the stabilisation and activity of the tumour suppressor p53 with siRNA targeting FAM83F reducing cellular proliferation (Salama et al, 2019). These effects are dependent on the mutational status of p53 with overexpression of FAM83F increasing cell migration in cells containing mutant p53, indicating FAM83F may promote or inhibit cancer progression depending on the tumour's mutational status. Recently, we have demonstrated that the membranous FAM83F–CK1α complex activates canonical Wnt signalling (Dunbar et al, 2020). Activated Wnt signalling is a hallmark of many cancers, especially colorectal cancers (CRCs) (Network, 2012). However, it is often believed that genetic alterations which activate canonical Wnt signalling in sporadic CRCs, specifically those at the level of the β-catenin destruction complex, will render any inhibition of upstream membranous Wnt signalling proteins ineffectual (Kahn, 2014). In contrast to this, we demonstrate that reducing membrane-associated FAM83F–CK1α with IMiD treatment can modulate canonical Wnt signalling in DLD-1 cells, which contain mutant Apc and are unresponsive to Wnt3A stimulation. Although there are benefits to testing the ability of IMiD-induced FAM83F degradation to attenuate Wnt signalling in cells containing Wnt activating mutations, the presence of truncated Apc adds increased complexity. Apc is a key component of the β-catenin destruction complex which regulates the canonical Wnt signalling pathway by regulating levels of the effector protein, β-catenin, and is required for Wnt signalosome formation after Wnt ligand binding (Clevers & Nusse, 2012; Parker & Neufeld, 2020). The presence of an Apc truncation mutation disrupts this complex to such an extent as to increase Wnt signalling but not completely abolish the complex, so that in DLD-1 cells, truncated Apc can still interact with other components of the β-catenin destruction complex and phosphorylation of β-catenin (S45) by CK1α is still present (Yang et al, 2006; Li et al, 2012). Therefore, there are caveats to be considered when evaluating Wnt signalling effects induced by removal of FAM83F–CK1α complexes in these cell lines. Lenalidomide has been tested in phase I and phase II clinical trials for sporadic CRCs but clinical response has been poor (Siena et al, 2013). Often these trials are in advanced metastatic tumours which contain multiple oncogenic driver mutations and aberrant signalling pathways. The clinical effect of IMiDs on an early Wnt-dependent disease such as the initial polyp formation in familial adenomatous polyposis, which is caused by mutant Apc (MacDonald et al, 1992), may yield more promising results.

# Materials and Methods

### Plasmids

All constructs are available for request from the Medical Research Council-Protein Phosphorylation and Ubiquitylation Unit (MRC-PPU) reagents website (http://mrcppureagents.dundee.ac.uk).

---

blotting with the indicated antibodies. **(D)** Densitometry of FAM83F and FAM83G protein abundance in anti-CK1α IP elute from (C). FAM83F and FAM83G protein abundances were normalised to CK1α protein abundance and represented as fold change compared with un-transfected HCT116 wild-type cells. Data presented as scatter graph illustrating individual data points with an overlay of the mean ± SD. **(E)** As in (C), except the cells were treated with IMiDs (10 μM for 24 h) as indicated and then were resolved by SDS–PAGE and subjected to Western blotting with the indicated antibodies. **(F)** Densitometry of FAM83F protein abundance upon treatment with IMiDs from (E). FAM83F protein abundance was normalised to GAPDH protein abundance and represented as fold change compared to untreated cells. Data presented as scatter graph illustrating individual data points with an overlay of the mean ± SD. **(B, F)** Statistical analysis of (B) and (F) data was completed using a Student's unpaired *t* test and comparing fold change between untreated and IMiD treated samples. Statistically significant *P*-values are denoted by asterisks (* < 0.05, ** < 0.01, *** < 0.001, **** < 0.0001).
Source data are available for this figure.

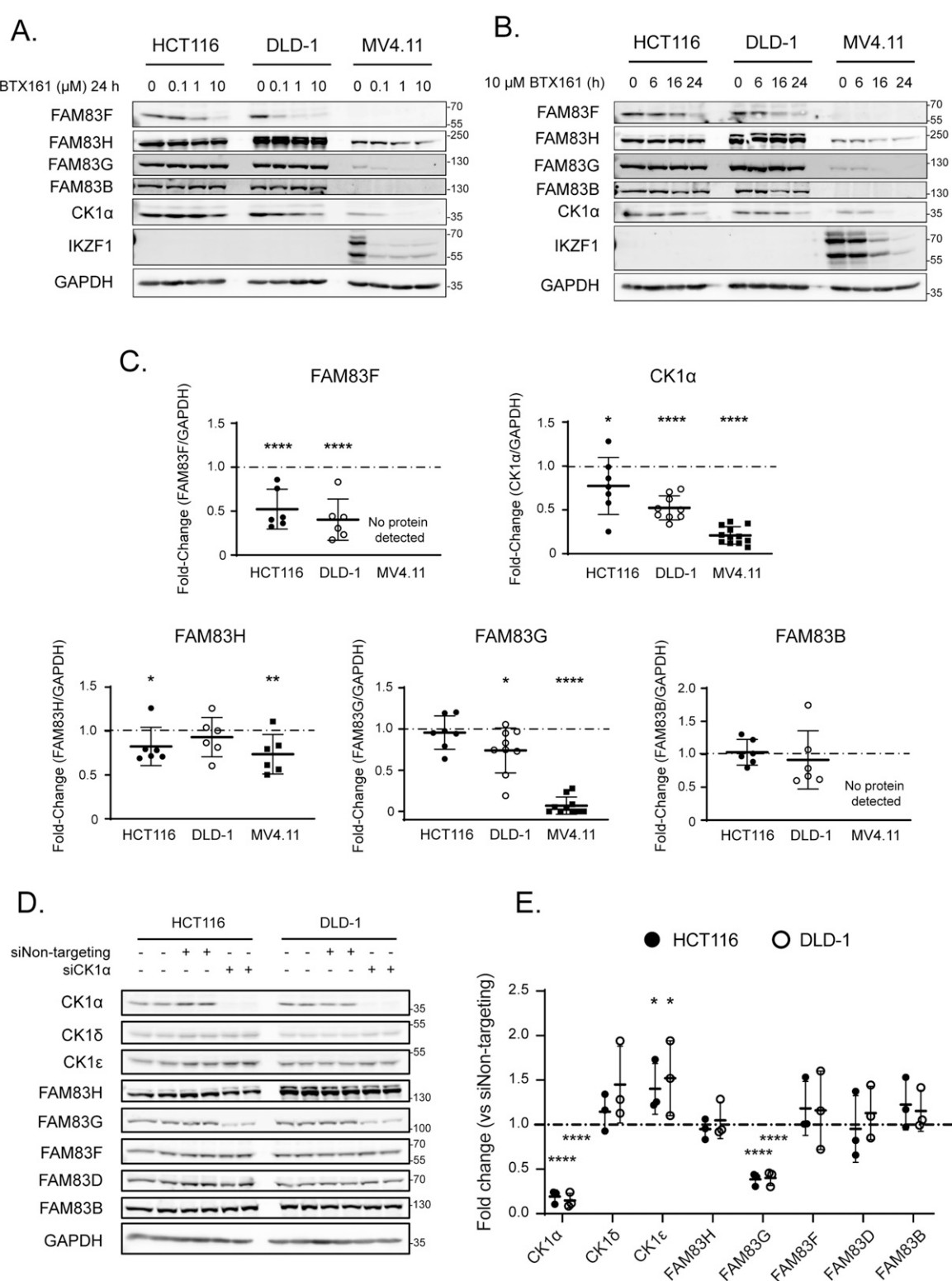

**Figure 6. BTX161 is an efficient CK1α-degrader which reduces FAM83G protein abundance through FAM83G-CK1α co-stability.**
**(A)** HCT116, DLD-1, and MV4.11 cell extracts treated with varying concentrations of BTX161 (0, 0.1, 1, or 10 μM) for 24 h were resolved by SDS–PAGE and subjected to Western blotting with the indicated antibodies. **(B)** As in (A) except cells were treated with 10 μM BTX161 for varying times (0, 6, 16 or 24 h) prior to lysis. **(C)** Densitometry of FAM83H, FAM83G, FAM83F, FAM83B, and CK1α protein abundance from (A, B) after treatment with 10 μM BTX161 for 24 h. Protein of interest abundance was normalised to GAPDH protein abundance and represented as fold change compared to untreated cells. Data presented as scatter graph illustrating individual data points with an overlay of the mean ± SD. The dashed line indicates a fold change of one which is equivalent to an untreated sample. Statistical analysis was completed using a Student's unpaired *t*

The unique identifier (DU) numbers provide direct links to the cloning strategy and sequence information. Sequences were verified by the DNA sequencing service, University of Dundee (http://www.dnaseq.co.uk). Constructs generated include pBABED.puro FAM83F (DU37979), pBABED.puro FAM83F$^{F284A/F288A}$ (DU28196), pBABED.puro FLAG CRBN (DU54685), pBABED.puro FLAG CRBN$^{V388I}$ (DU64137), pBABED.puro U6 FAM83F tv1 Nter KI sense (DU54050), pX335 FAM83F Nter KI Antisense (DU54056), pMS-RQ FAM83F Nter GFP donor (DU54325), pBABED.puro U6 FAM83F ex2 KO sense (DU54848), pX335 FAM83F ex2 KO Antisense (DU54850), pBABED.puro U6 CRBN ex3 KO sense A (DU64046), pX335 CRBN ex3 KO Antisense A (DU64483), pBABED.puro U6 FAM83G ex2 KO sense (DU52480), pX335 FAM83G ex2 KO Antisense (DU52484), pcDNA5-FRT/TO-GFP-FAM83G (DU33272), and pcDNA5-FRT/TO-GFP-FAM83G$^{F296A}$ (DU28477).

Plasmid amplification was completed by transforming 10 μl *Escherichia coli* DH5α competent cells (Invitrogen) using 1 μl of plasmid DNA. Bacteria were incubated on ice for 10 min before heat-shocking at 42°C for 45 s. After a further 2 min on ice, the transformed bacteria were plated on LB agar medium plates containing 100 μg/ml ampicillin and incubated at 37°C for 16 h. Single colonies were used to inoculate a 5 ml culture of LB medium containing 100 μg/ml ampicillin then incubated at 37°C for 16 h with constant shaking. Plasmid DNA was purified from a bacterial culture using QIAGEN Miniprep Kit by following the manufacturer's protocol. Isolated DNA yield was quantified using a Nano-Drop 1000 spectrophotometer (Thermo Fisher Scientific).

## Antibodies

Antibodies recognising FAM83B (SA270), FAM83D (SA102), FAM83F (SA103), FAM83H (SA273), CK1α (SA527), CK1ε (SA610), CK1δ (SA609), and GFP (S268B) were generated in-house and are available for request from the MRC-PPU reagents website (http://mrcppureagents.dundee.ac.uk). Antibodies recognising GAPDH (14C10) (#2118), IKZF1 (D6N9Y) (#14859), CRBN (D8H3S) (#71810), Na, K-ATPase alpha1 (D4Y7E) (#23565), and Lamin A/C (#2032) were obtained from Cell Signalling Technology. Additional antibodies used were FAM83G (ab121750; Abcam), α-tubulin (MA1-80189; Thermo Fisher Scientific), Ubiquitin (BML-PW8810; Enzo), HIF-1α (6109590; BD Biosciences), and ZFP91 (A303-245A; Bethyl Laboratories). Secondary antibodies used were Star-Bright Blue 700 goat anti-rabbit IgG (12004161; Bio-Rad), StarBright Blue 700 goat anti-mouse IgG (12004158; Bio-Rad), IRDye 800CW donkey anti-goat IgG (926-32214; LI-COR) and IRDye 800CW goat anti-rat IgG (926-32219; LI-COR).

## Cell culture

THP-1 (TIB-202; ATCC) and MV4.11 (CRL-9591; ATCC) cells were maintained in Roswell Park Memorial Institute 1640 medium (RPMI; Gibco). HCT116 (CCL-247; ATCC), DLD-1 (CCL-221; ATCC), PC-3 (CRL-1435; ATCC), A549 (CCL-185; ATCC), U2OS (HTB-96; ATCC), HEK-293 (CRL-1573; ATCC), ARPE-19 (CRL-2302; ATCC), SH-SY5Y (CRL-2266; ATCC), G-361 (CRL-1424; ATCC), SK-MEL-13 (RRID:CVCL_6022), HaCaT (from Joan Massague's lab at Memorial Sloan Kettering Cancer Centre, not commercially obtained but can be provided on request) (Sapkota et al, 2007), and HEK-293-FT (R70007; Thermo Fisher Scientific) cells were maintained in DMEM (Gibco). RPMI and DMEM were supplemented with 10% (vol/vol) FBS (F7524; Sigma-Aldrich), 2 mM L-glutamine (25030024; Invitrogen), 100 U/ml penicillin, and 100 mg/ml streptomycin (15140122; Invitrogen). Cells lines were regularly tested for mycoplasma contamination and only mycoplasma-free cell lines were used for experimentation.

## Generation of $^{GFP/GFP}$FAM83F, FAM83F$^{-/-}$, CRBN$^{-/-}$, and FAM83G$^{-/-}$ cell lines using CRISPR/Cas9 genome editing

All CRISPR/Cas9 targeting procedures were performed using a dual guide nickase strategy. For the generation of FAM83F knockout HCT116 and DLD-1 cell lines, the *FAM83F* locus was targeted with sense guide RNA (pBabeD-puro vector, DU54848); GCGTCCAGGATGATGTACACT and antisense guide RNA (pX335-Cas9-D10A vector, DU54850); GGCAGGAGT-GAAGTATTTCC. For the generation of CRBN knockout DLD-1 cell lines, the *CRBN* locus was targeted with sense guide RNA (pBabeD-puro vector, DU64046); GCTCAAGAAGTCAGTATGGTG and antisense guide RNA (pX335-Cas9-D10A vector, DU64483); GTGAAGAGGTAATGTCTGTCC. For the generation of FAM83G knockout HCT116 cell lines, the *FAM83G* locus was targeted with sense guide RNA (pBabeD-puro vector, DU52480); GGACCGCTCCATCCCGCAGC and antisense guide RNA (pX335-Cas9-D10A vector, DU52484); GCTGGGGCCAGTACTCCAGGG. For generation of N-terminal GFP knock-in to the *FAM83F* locus, the *FAM83F* locus was targeted with sense guide RNA (pBabeD-puro vector, DU54050); GTTCAGCTGGGACTCGGCCA, antisense guide RNA (pX335-Cas9-D10A vector, DU54056); GCGAGGCGCACGTGAACGAGA and the GFP-FAM83F donor (pMK-RQ vector, DU54325).

Plasmids (1 μg of sense and antisense guide RNAs + 3 μg donor for knock-ins) were diluted in 1 ml OptiMem (Gibco) and 20 μl of polyethylenimine (PEI; 1 mg/ml) (Polysciences) was added. This transfection mix was vortexed vigorously for 15 s, incubated for 20 min at room temperature and then added dropwise to a 10-cm diameter dish containing ~70% confluent cells in complete culture medium. Selection of transfected cells was performed 24 h post transfection in medium containing 2 μg/ml puromycin for 48 h. Single cells were isolated by FACS, with single GFP-positive cells (for knock-ins) or all cells (for knockouts) isolated, and plated into individual wells of 96-well plates, pre-coated with 1% (wt/vol) gelatin (Sigma-Aldrich). Viable clones were expanded and assessed for successful knock-in or knockout by both Western blotting and genomic DNA sequencing (Figs S3–S5).

For verification by DNA sequencing, the region surrounding the gRNA target sites were amplified by PCR with KOD Hot Start Polymerase (Merck) according to manufacturer's instructions with the following

---

test and comparing fold change between untreated and samples treated with 10 μM BTX161 for 24 h. **(D)** HCT116 and DLD1 cell extracts transfected with control siRNA or siCK1α for 48 h were resolved by SDS–PAGE and subjected to Western blotting with the indicated antibodies. **(E)** Densitometry of CK1α, CK1δ, CK1ε, FAM83H, FAM83G, FAM83F, FAM83D and FAM83B protein abundance from (D) normalised to GAPDH protein abundance and represented as fold change compared to HCT116 and DLD-1 cells transfected with Non-targeting siRNA control. Data presented as scatter graph illustrating individual data points with an overlay of the mean ± SD. The dashed line indicates a fold change of one which is equivalent to an untreated sample. Statistical analysis was completed using a Student's unpaired *t* test and by comparing fold change between cells transfected with Non-targeting siRNA control and cells transfected with siCK1α. Statistically significant *P*-values are denoted by asterisks (* < 0.05, ** < 0.01, *** < 0.001, **** < 0.0001).
Source data are available for this figure.

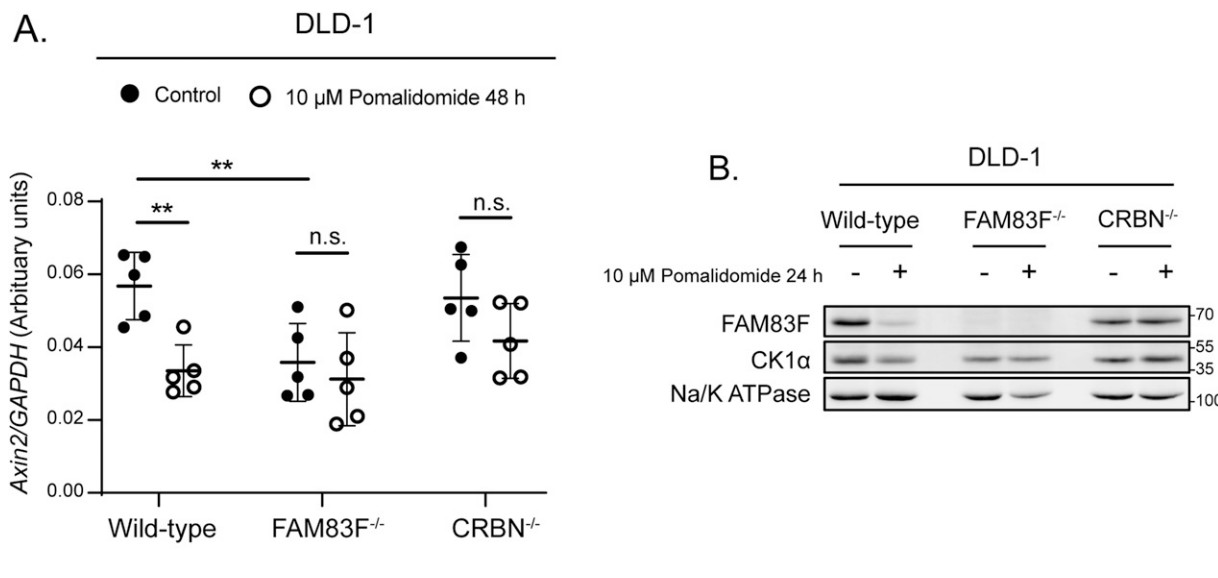

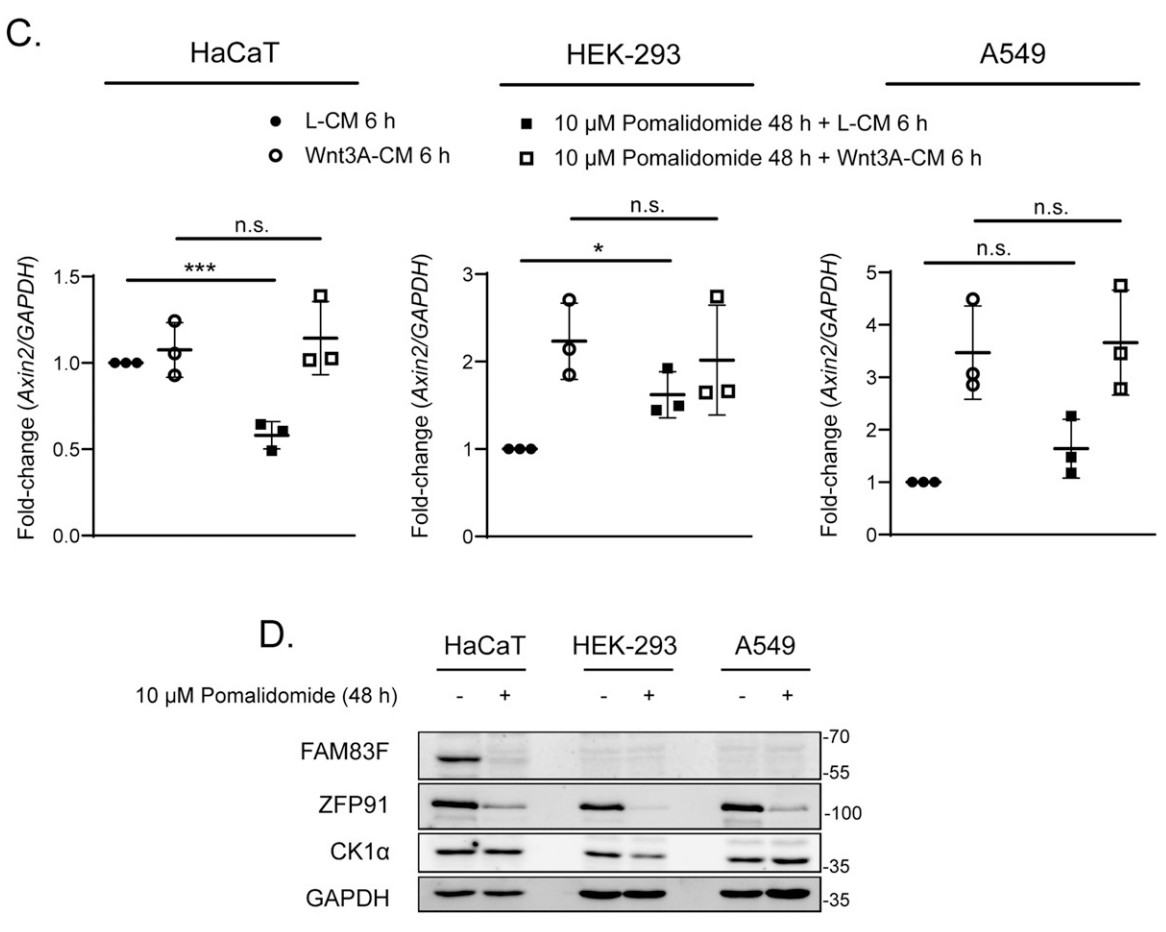

**Figure 7. IMiD-induced degradation of FAM83F attenuates Wnt signalling and removes CK1α from the plasma membrane.**
**(A)** qRT-PCR was performed using cDNA from DLD-1 wild-type, DLD-1 FAM83F[−/−] and CRBN[−/−] cell lines following treatment with 10 μM pomalidomide for 48 h, and primers for *Axin2* and *GAPDH* genes. *Axin2* mRNA expression was normalised to *GAPDH* mRNA expression and represented as arbitrary units. Data presented as scatter graph illustrating individual data points with an overlay of the mean ± SD. **(B)** Membrane fractions from DLD-1 wild-type, DLD-1 FAM83F[−/−], and DLD-1 CRBN[−/−] cell lines, following treatment with 10 μM pomalidomide for 24 h, were resolved by SDS–PAGE and subjected to Western blotting with the indicated antibodies. The specificity of membrane compartment isolation was determined with Western blotting for Na/K ATPase, a membrane specific protein. **(C)** qRT-PCR was performed using cDNA derived

A. Lenalidomide and BTX161 compounds can induce degradation of free CK1α

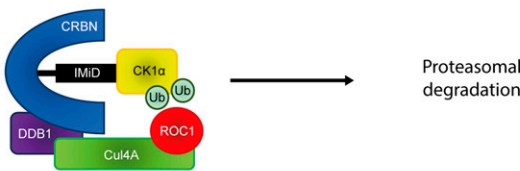

Proteasomal degradation

B. FAM83F degradation by IMiD compounds requires CK1α-binding and CRBN

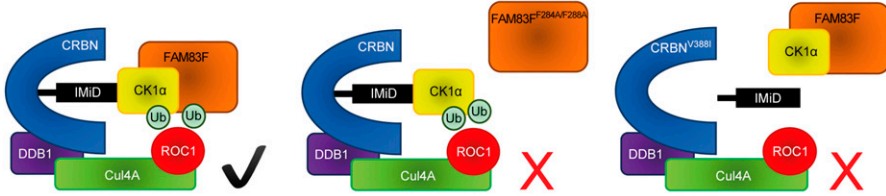

C. BTX161 can effciently degrade CK1α in MV4.11 cells leading to reduced FAM83G protein abundance

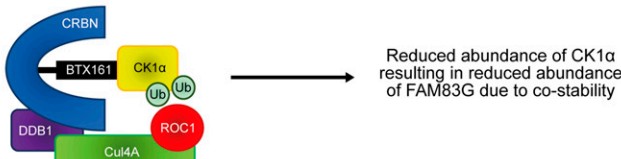

Reduced abundance of CK1α resulting in reduced abundance of FAM83G due to co-stability

**Figure 8. Proposed model for IMiD-induced FAM83F and CK1α degradation.**
**(A)** Previously reported mode of action for CK1α degradation by lenalidomide and BTX161 compounds. The IMiD can bind CK1α and a conserved binding pocket in CRBN, thus bringing CK1α into proximity of the Cul4A$^{CRBN}$ complex, facilitating the addition of ubiquitin by ROC1 to CK1α. CK1α is then subsequently degraded via the proteasome. **(B)** Our proposed model for FAM83F degradation by multiple IMiDs. The IMiDs can bind the CK1α–FAM83F complex and a conserved binding pocket in CRBN, thus bringing the CK1α–FAM83F complex into proximity of the Cul4A$^{CRBN}$ complex, facilitating the addition of ubiquitin by ROC1 to the CK1α–FAM83F complex. CK1α and FAM83F are then subsequently degraded via the proteasome. Mutation of FAM83F at F284 and F288 to alanines abolishes the interaction with CK1α, thus there is no recruitment or subsequent degradation of FAM83F upon IMiD treatment. Mutation of CRBN at V388 to isoleucine to mimic the mouse homolog which cannot bind IMiDs prevents IMiD from binding to CRBN, thus no IMiD neo-substrate, including the CK1α–FAM83F complex, can be recruited to the Cul4A$^{CRBN}$ complex. **(C)** We propose that the BTX161-induced reduction in FAM83G protein abundance is a result of efficient CK1α degradation by BTX161 resulting in a loss of FAM83G protein due to FAM83G-CK1α co-stability, and not direct BTX161-induced FAM83G degradation.

primer pairs: FAM83F exon 2 (Forward: TCATTGCTGTGGTCATGGAC, Reverse: AATCCGGAAGTCAGTGAGCT), FAM83F N-terminal GFP KI (Forward: TGCGCGGAAAATGAACTCGTACC, Reverse: GTAGAAACCAGTGTCCGTCCAGC), CRBN exon 3 (Forward: GGTGCTGATATGGAAGAATTTCATGGC, Reverse: GTATGAAGGTGAAGAGCTGAGTTAGATGG) and FAM83G exon 2 (Forward: TCTTTCCCGCAGATTGCTCATGG, Reverse: TTCTTCTGGGGAACCAGAAACACC). PCR products of positive clones were cloned with the StrataClone PCR Cloning Kit (Agilent) into the supplied vector system, according to the manufacturer's protocol. Sequencing of the edited loci in the positive clones was performed by the MRC-PPU DNA sequencing and services (http://mrcppureagents.dundee.ac.uk).

**Transient transfections**

Transient transfections were performed in HCT116 cells with either pcDNA5-FRT/TO-GFP-FAM83G (DU33272) or pcDNA5-FRT/TO-GFP-FAM83G$^{F296A}$ (DU28477). Plasmids (1 μg) were diluted in 1 ml OptiMem

(Gibco) and 20 μl PEI (1 mg/ml) was added. The transfection mix was incubated for 20 min at room temperature, then added dropwise to a 10-cm$^2$ diameter dish of cells in complete culture medium. Fresh media was added 24 h post-transfection and indicated treatments performed thereafter.

**Retroviral transductions**

Retroviruses were produced using the following constructs: pBA-BED.puro FAM83F$^{WT}$ (DU37979), pBABED.puro FAM83F$^{F284A/F288A}$ (DU28196), pBABED.puro FLAG CRBN (DU54685), and pBABED.puro FLAG CRBN$^{V388I}$ (DU64137). Retroviruses were produced by transfecting HEK-293-FT cells as previously described (Fulcher et al, 2019). Briefly, 6 μg pBabe plasmid, 3.8 μg pCMV5-GAG/Pol (Clontech), and 2.2 μg pCMV5-VSV-G (Clontech) were diluted in 600 μl OptiMem (Gibco) and 24 μl PEI (1 mg/ml) was added. The transfection mixture was incubated for 20 min at room temperature then added dropwise to a 10-cm diameter dish of cells in

from HaCaT, HEK-293 and A549 wild-type cell extracts following treatment with 10 μM pomalidomide for 48 h and a further treatment with either L-CM or Wnt3A-CM for 6 h before lysis, and primers for *Axin2* and *GAPDH* genes. *Axin2* mRNA expression was normalised to *GAPDH* mRNA expression and is represented as fold change compared with untreated cells. Data presented as scatter graph illustrating individual data points with an overlay of the mean ± SD. **(D)** Extracts derived from HaCaT, HEK-293 and A549 wild-type cells treated with 10 μM pomalidomide for 48 h were resolved by SDS–PAGE and subjected to Western blotting with indicated antibodies. Statistical analysis of data was completed using a Student's unpaired *t* test. Statistically significant *P*-values are denoted by asterisks (* < 0.05, ** < 0.01, *** < 0.001, **** < 0.0001). n.s., non-significant.
Source data are available for this figure.

complete culture medium. Fresh media was added 24 h post-transfection. Media containing retroviruses was collected after 24 h and passed through a 0.45 $\mu$m sterile syringe filter. For transduction of target cells, 1 ml of retroviral medium together with 8 $\mu$g/ml polybrene (Sigma-Aldrich) was added to a 10 cm diameter dish of cells containing 9 ml complete culture medium. Selection of transduced cells was performed 24 h post transduction with cells incubated in media containing 2 $\mu$g/ml puromycin for 48 h. Successful transduction was assessed by Western blotting.

### Compound treatments

IMiD compounds (Thalidomide, Lenalidomide, Pomalidomide and Iberdomide) were obtained from Caymen Chemicals. BTX161 compound was synthesised in-house and is available for request from the MRC-PPU reagents website (http://mrcppureagents.dundee.ac.uk). IMiD compounds were added to cell culture media at indicated concentrations (between 0.1 and 10 $\mu$M) for indicated duration. dTAG-13 (Sigma-Aldrich), a CRBN-binding PROTAC, was used at 1 $\mu$M for 24 h. MLN4924 (Sigma-Aldrich), an inhibitor of NEDD8-activating E1 enzyme, was used at 1 $\mu$M for 24 h. Bortezomib (Sigma-Aldrich), a proteasome inhibitor, was used at 5 $\mu$M for 24 h.

### Generation of L- and Wnt3A-conditioned medium (CM)

Mouse fibroblast L-cells (CRL-2648; ATCC) and mouse fibroblast L-cells that stably overexpress Wnt3A (CRL-2647; ATCC) were maintained in DMEM containing 10% FCS. To generate CM, L-cells and L-Wnt3A cells were grown in DMEM in 15-cm diameter dishes for 3 d. Medium was collected, filtered (0.22 $\mu$m) and stored as L-CM and Wnt3A-CM (Wnt3A-CM). Before treatment, CM was diluted 50:50 in DMEM containing 10% FCS.

### Cell lysis, SDS–PAGE, and Western blotting

Cells were washed and scraped in ice-cold PBS, and then pelleted. For whole cell lysates, cell pellets were lysed in lysis buffer (20 mM Tris–HCl [pH 7.5], 150 mM NaCl, 1 mM EDTA, 1 mM EGTA, 1% [vol/vol] Triton X-100, 2.5 mM sodium pyrophosphate, 1 mM $\beta$-glycerophosphate, 1 mM Na$_3$VO$_4$, and 1× complete EDTA-free protease inhibitor cocktail [Roche]). Lysates were clarified at 17,000$g$ for 20 min. Cellular fractionation into cytoplasmic, nuclear and membrane lysates was completed using a subcellular protein fractionation kit (Thermo Fisher Scientific) following the manufacturer's instructions. Briefly, cells were lysed in sequential buffers to separate cellular compartments into cytoplasmic, membrane, nuclear, and cytoskeletal fractions. Protein concentration was measured using Pierce Coomassie Bradford Protein Assay Kit (Thermo Fisher Scientific). Protein concentrations were adjusted to 1–3 $\mu$g/$\mu$l in lysis buffer and NuPAGE 4× LDS sample buffer (NP0007) (Thermo Fisher Scientific) was added to lysates. Lysates (15–30 $\mu$g protein) were separated by sodium dodecyl sulphate-polyacrylamide electrophoresis (SDS–PAGE) and gels were transferred to nitrocellulose membranes. After blocking in 5% (wt/vol) milk in TBS-T (50 mM Tris–HCL [pH 7.5], 150 mM NaCl, 0.1% [vol/vol] Tween-20) for 60 min, membranes were incubated in primary antibody (1:1,000 dilution in blocking buffer) for 16 h at 4°C, washed 3 × 10 min in TBS-T, then incubated in secondary antibody (1:5,000 dilution in blocking buffer) for 1 h at room temperature and washed

3 × 10 min in TBS-T. Fluorescence of secondary antibody was detected using the Chemidoc system (Bio-Rad) and data analysed by Image lab software (Bio-Rad). Densitometry of protein blots was completed using Image J software (https://imagej.net). The density of protein of interest bands were measured and normalised to those of loading control bands with fold-change calculations and statistical analysis performed using Microsoft Excel software (www.microsoft.com). Graphical representations of data were prepared using Prism 8 (www.graphpad.com).

### Immunoprecipitation

Protein lysates were prepared, and protein concentration quantified as previously outlined. Anti-CK1$\alpha$ antibody (1 $\mu$g) was added to each lysate sample (1 mg protein) and incubated on a rotating wheel for 16 h at 4°C. Protein G sepharose beads (DSTT) pre-equilibrated in cell lysis buffer were added (20 $\mu$l of 50% beads: lysis buffer slurry) to each lysate sample and incubated on a rotating wheel for 1 h at 4°C. Beads were pelleted and supernatant removed and stored as flow-through. Beads were washed in cell lysis buffer three times. Elution was performed by the addition of 40 $\mu$l of NuPAGE 1× LDS sample buffer to the beads, followed by denaturing proteins at 95°C for 10 min. Input and eluted samples were analysed by SDS–PAGE as previously described.

### Immunofluorescence

Cells were plated on sterile glass coverslips. Cells were fixed in 4% (vol/vol) paraformaldehyde for 15 min, permeabilized in 0.2% (vol/vol) Triton X-100 for 10 min, and then blocked in 5% (wt/vol) bovine serum albumin for 60 min. Cells were incubated in primary antibody (1:100) diluted in 0.5% (wt/vol) bovine serum albumin for 16 h at 4°C. After washing in PBS/0.1% Tween-20, cells were incubated in Alexa Fluor 594 secondary (1:500; Thermo Fisher Scientific) for 60 min at room temperature. Coverslips were washed in PBS/0.1% Tween-20 and then incubated with 1 $\mu$g/ml DAPI (Sigma-Aldrich) for 5 min at room temperature to visualise nuclei. Coverslips were mounted on glass slides in Vectashield mounting media (Vector Laboratories) and sealed with CoverGrip coverslip sealant (Biotium). Images were captured on a DeltaVision microscope using 60× objectives. Images were prepared using Omero software (www.openmicroscopy.org).

### Transfection with siRNA

ON-TARGETplus human *CSNK1A1* siRNA (L-003957-00-0005; Dharmacon) and ON-TARGETplus Non-targeting pool (D-001810-10-05; Dharmacon) were resuspended using 5× siRNA buffer (B-002000-UB-100; Dharmacon). Adherent cells were seeded in six-well plates and grown to 70–80% confluence. siRNA was diluted in OptiMem (Gibco) to a final working concentration of 25 pmol siRNA per well of a six-well plate. Lipofectamine RNAi-MAX transfection reagent (13778100; Thermo Fisher Scientific) was diluted in OptiMem to a final working volume of 7.5 $\mu$l Lipofectamine per well. After incubating both siRNA and Lipofectamine samples separately for 5 min at room temperature, they were mixed and incubated for 20 min at room temperature. The siRNA-Lipofectamine mixture was added dropwise to cells and incubated for 48 h before lysis.

## Quantitative real-time PCR (qRT-PCR)

Cells were grown to ~70% confluence in six-well plates. Cells were then treated with IMiD compounds as indicated before RNA extractions were performed using the RNeasy Mini Kit (QIAGEN). RNA was quantified using a NanoDrop 3300 Fluorospectrometer (Thermo Fisher Scientific). Synthesis of cDNA was completed using iScript cDNA Synthesis Kit (Bio-Rad) and 1 $\mu$g of RNA per reaction. qRT-PCR was performed in triplicate in a 10 $\mu$l final volume with 2 $\mu M$ forward primer, 2 $\mu M$ reverse primer, 50% (vol/vol) iQ SYBR Green Supermix (Bio-Rad), and 2 $\mu$l cDNA (diluted 1:5) using a CFX384 real-time system qRT-PCR machine (Bio-Rad). Primers were designed using Benchling and purchased from Invitrogen. *Axin2* forward: TACACTCCTTATTGGGCGATCA, *Axin2* reverse: TTGGCTACTCGTAAAGTTTTGGT, *GAPDH* forward: TGCACCACCAACTGCTTAGC, *GAPDH* reverse: GGCATG-GACTGTGGTCATGAG. The datasets were analysed using the comparative Ct method (ΔΔCt Method) (Livak & Schmittgen, 2001) with *Axin2* the Wnt-target gene and *GAPDH* the endogenous control gene. Statistical analysis was performed using Microsoft Excel software (www.microsoft.com) and graphical representations of data were prepared using Prism 8 (www.graphpad.com).

## Supplementary Information

## Acknowledgements

We thank E Allen, L Fin, J Stark, and A Muir for help and assistance with tissue culture, the staff at the DNA sequencing service (School of Life Sciences, University of Dundee), the cloning, antibody, and protein production teams within the Medical Research Council-Protein Phosphorylation and Ubiquitylation Unit (MRC-PPU) reagents and services (University of Dundee) coordinated by J Hastie and Natalia Shpiro (MRC-PPU) for synthesis of BTX161. We thank the staff at the Dundee Imaging Facility (School of Life Sciences, University of Dundee) and the Flow Cytometry Facility (School of Life Sciences, University of Dundee) for their invaluable help and advice throughout this project. We thank all members of the Sapkota lab for their highly appreciated experimental advice and/or discussions. Funding: K Dunbar is supported by an MRC Career Development Fellowship. GP Sapkota is supported by the UK Medical Research Council (grant numbers MC_UU_00018/6 and MC_UU_12016/3) and the pharmaceutical companies supporting the Division of Signal Transduction Therapy (DSTT) (Boehringer-Ingelheim, GlaxoSmithKline, Merck-Serono).

### Author Contributions

K Dunbar: data curation, formal analysis, investigation, and writing—original draft, review, and editing.
TJ Macartney: CRISPR strategy, design, methodology, and cloning.
GP Sapkota: conceptualization, formal analysis, supervision, funding acquisition, project administration, and writing—review and editing.

### Conflict of Interest Statement

The authors declare that they have no conflict of interest.

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
