## [Reviewer comments · Life Science Alliance]

Life Science Alliance

IMiDs induce FAM83F degradation via an interaction with CK1 α to attenuate Wnt signalling

Gopal Sapkota, Karen Dunbar, and Thomas Macartney

DOI: <https://doi.org/10.26508/lsa.202000804>

Corresponding author(s): Gopal Sapkota, University of Dundee

Review Timeline:	Submission Date:	2020-06-01
	Editorial Decision:	2020-07-31
	Revision Received:	2020-10-30
	Editorial Decision:	2020-12-08
	Revision Received:	2020-12-11
	Accepted:	2020-12-11

Scientific Editor: Shachi Bhatt

Transaction Report:

July 31, 2020

Re: Life Science Alliance manuscript #LSA-2020-00804-T

Dr. Gopal P. Sapkota
University of Dundee
MRC Protein Phosphorylation and Ubiquitylation Unit
School of Life Sciences
Dow Street
Dundee, Scotland DD1 5EH
United Kingdom

Dear Dr. Sapkota,

Thank you for submitting your manuscript entitled "IMiDs induce FAM83F degradation via an interaction with CK1 α to attenuate Wnt signalling" to Life Science Alliance, which was now assessed by two referees, whose reports are copied below.

Referees appreciate the analysis, but they also raise significant concerns that need to be addressed for publication in this journal. For example, referees would like an explanation as to why FAM83F is selectively recognized by IMiDs and they raise concerns regarding data presentation and statistical analyses. We find the reports constructive and informed, and believe that addressing referee concerns will strengthen the manuscript.

Given these positive recommendations, we would like to invite you to revise your manuscript with the understanding that the referee concerns (as in their reports) must be fully addressed and their suggestions taken on board. Please address all referee concerns in a complete point-by-point response. Acceptance of the manuscript will depend on a positive outcome of a second round of review, and we require strong support from referees for publication here.

In our view these revisions should typically be achievable in around 3 months. However, we are aware that many laboratories cannot function fully during the current COVID-19/SARS-CoV-2 pandemic and therefore encourage you to take the time necessary to revise the manuscript to the extent requested above. We will extend our 'scoping protection policy' to the full revision period required. If you do see another paper with related content published elsewhere, nonetheless contact me immediately so that we can discuss the best way to proceed.

Thank you for this interesting contribution to Life Science Alliance. We are looking forward to receiving your revised manuscript.

Sincerely,

Reilly Lorenz
Editorial Office Life Science Alliance
Meyerhofstr. 1
69117 Heidelberg, Germany
t +49 6221 8891 414
e contact@life-science-alliance.org
www.life-science-alliance.org

B. MANUSCRIPT ORGANIZATION AND FORMATTING:

Reviewer #2 (Comments to the Authors (Required)):

Casein Kinase 1 alpha (CK1 α) is a known substrate of the thalidomide immunomodulatory imide drugs that recruit neo-substrates to the cereblon-containing ubiquitin ligase. Since the authors have found that FAM83 family members interacts strongly with CK1 α , here they investigate if the IMiDs influence FAM83 protein stability. They find FAM83F is a target, perhaps even better than CK1 α , and they demonstrate that IMiD-induced degradation of FAM83F requires interaction with CK1 α . Conversely, FAM83G protects CK1 α from degradation. The studies are well controlled, the manuscript is well written, and the conclusions are supported by the data. This is a valuable contribution to our understanding of the function of IMiDs and the regulation of CK1 α and the FAM83 family.

My only suggestion is again, plunger plots should be replaced with scatter plots to allow the reader to see the data.

Reviewer #3 (Comments to the Authors (Required)):

IMiDs induce FAM83F degradation via an interaction with CK1 α to attenuate Wnt Signalling

This manuscript deals with the property of IMiDs to bind to CRBN, a substrate for the ubiquitination machinery (more specifically Cul4A), and, by binding to other cellular components, recruit them a "neo-substrates" and drive their degradation.

Building on knowledge that IMiD drugs can lead to casein kinase alpha ubiquitination, and that the FAM83 family binds CK1alpha, the authors identify FMA83F as neo-substrate for two IMiD drugs. The authors show in a second submitted manuscript that FAM83F is a potentiator of Wnt signalling through its ability to sequester CK1a. Thus, its IMiD-induced degradation could potentially downregulate Wnt signalling in cancer. Yet, CK1a is in fact poorly degraded in response to IMiD treatment in most cancer cell lines. The authors propose that this is due to the protective action of another member of the family, FAM83G, which is not targeted by IMiDs. More generally, they propose a model where IMiD efficiency at inducing degradation of CK1a depends on the type and level FAM83 proteins expressed. Thus, the high efficiency in multiple myeloma, which express only traces of these proteins.

While the study is in most aspects well done and coherent, I have a few comments and queries.

1) FAM83F is supposed to be recognized indirectly by IMiD, via CK1a. I then don't understand why FAM83G would not be recognized. It would be unfortunate if this result may be due to other parameters (expression levels? Differences in accessibility of the FAM83-CK1a complex?). This seems to be an important issue, since these parameters may be context-dependent, and thus FAM83G may well be degraded in other cell types. The same goes obviously for other members of the family.

2) In my view, one of the most striking observations of this study is the clear difference between the

poor degradation of CK1a in most cell lines (except MV4.11), in comparison to the very strong degradation of FAM83F. These results should be more clearly highlighted and discussed.

3) One serious issue: Stats are absent from most blot quantifications, including those from Fig.1 on which the whole work is based. Without proper statistical analysis, these quantifications have little value.

4) Along the same lines, in those panels where stats are included, one sees p values of 0.08 and 0.058, as well as several cases with a single * ($p < 0.05$). Although it is completely fine to show such values, it is very dangerous to extract conclusions out of such data. There is very good literature explaining how "weak" is the significance of a $p = 0.05$.

5) The choice of DLD-1 cells, which have a mutated APC, as model to test the impact on Wnt pathway is interesting for potential therapeutic applications, but biologically, the rationale is unclear. One would wish a) a detailed rationale which will take into account the known composition/function of the Axin-APC complex and of the position of CK1a in this context. b) Verification of the effect on Wnt signalling in a cell line with a functional pathway.

6) Figure 2: The IF signals are very weak, in particular for CK1 α . Please increase contrast and perhaps show the CK1a channel in gray scale for better visualization.

Responses to reviewer's comments: The reviewer's comments are *italicised*, and our responses appear as non-italicised fonts. New data and figures are indicated with bold face fonts.

LSA-2020-00804-T: IMiDs induce FAM83F degradation via an interaction with CK1 α to attenuate Wnt signalling

Reviewer #2

Casein Kinase 1 alpha (CK1 α) is a known substrate of the thalidomide immunomodulatory imide drugs that recruit neo-substrates to the cereblon-containing ubiquitin ligase. Since the authors have found that FAM83 family members interacts strongly with CK1 α , here they investigate if the IMiDs influence FAM83 protein stability. They find FAM83F is a target, perhaps even better than CK1 α , and they demonstrate that IMiD-induced degradation of FAM83F requires interaction with CK1 α . Conversely, FAM83G protects CK1 α from degradation. The studies are well controlled, the manuscript is well written, and the conclusions are supported by the data. This is a valuable contribution to our understanding of the function of IMiDs and the regulation of CK1 α and the FAM83 family.

My only suggestion is again, plunger plots should be replaced with scatter plots to allow the reader to see the data.

Response: We thank the reviewer for the positive review of our manuscript. We have taken the reviewer's suggestion on board and replaced all plunger plots with scatter plots, which illustrate all individual data points with an overlay of the mean and standard deviation.

Reviewer #3

This manuscript deals with the property of IMiDs to bind to CRBN, a substrate for the ubiquitination machinery (more specifically Cul4A), and, by binding to other cellular components, recruit them a "neo-substrates" and drive their degradation.

Building on knowledge that IMiD drugs can lead to casein kinase alpha ubiquitination, and that the FAM83 family binds CK1alpha, the authors identify FAM83F as neo-substrate for two IMiD drugs. The authors show in a second submitted manuscript that FAM83F is a potentiator of Wnt signalling through its ability to sequester CK1a. Thus, its IMiD-induced degradation could potentially downregulate Wnt signalling in cancer. Yet, CK1a is in fact poorly degraded in response to IMiD treatment in most cancer cell lines. The authors propose that this is due to the protective action of another member of the family, FAM83G, which is not targeted by IMiDs. More generally, they propose a model where IMiD efficiency at inducing degradation of CK1a depends on the type and level FAM83 proteins expressed. Thus, the high efficiency in multiple myeloma, which express only traces of these proteins.

While the study is in most aspects well done and coherent, I have a few comments and queries.

1) FAM83F is supposed to be recognized indirectly by IMiD, via CK1a. I then don't understand why FAM83G would not be recognized. It would be unfortunate if this result may be due to other parameters (expression levels? Differences in accessibility of the FAM83-

CK1a complex?). This seems to be an important issue, since these parameters may be context-dependent, and thus FAM83G may well be degraded in other cell types. The same goes obviously for other members of the family.

Response: We thank the reviewer for an in-depth and constructive appraisal of our manuscript. To address the potential cell line dependency of IMiD-induced FAM83 degradation, we have added data from a further six cell lines (**Supplementary Figure 1**) to complement the previously tested six cell lines (Figure 1C) and except for FAM83F, we did not identify any other FAM83 protein degraded by the IMiD compounds. Additionally, we have confirmed that the interaction between FAM83G and CK1 α is unaffected by IMiD treatment (**Response Figure 1**). We performed FAM83G immunoprecipitations in HCT116 wild-type and FAM83G^{-/-} cells following pomalidomide treatment and observed no change in the interaction between FAM83G and CK1 α . Therefore, we have no evidence that FAM83G can be degraded by IMiD compounds despite a robust FAM83G-CK1 α interaction. Based on our data we cannot make any assumptions regarding the potential IMiD-induced degradation of FAM83A, FAM83C or FAM83E, as we have yet to identify robust antibodies which recognise these proteins. However, a report from Donovan K.A. et al that detailed a comprehensive proteomic screen to identify novel targets of IMiD-inducible protein degradation in multiple cell lines including human embryonic stem cells, neuroblastoma and multiple myeloma derived cell lines [1], found a significant reduction of FAM83F protein abundance after treatment with multiple IMiD compounds but no other FAM83 protein was shown to be degraded upon IMiD treatment in these cell lines, suggesting that other FAM83 proteins are spared from degradation. We have now referenced this study in the discussion section with the following text: “The specific degradation of FAM83F and the absence of degradation of other FAM83 proteins, after treatment with IMiD compounds, has been corroborated by quantitative mass spectrometry [1].” Whether all FAM83 proteins, especially FAM83G, are recruited to CRBN upon IMiD treatment through CK1 α but are prevented from ubiquitylation possibly due to inaccessible surface Lys residues resulting from the nature of the specific FAM83-CK1 α complex remains unresolved but could be likely. On the other hand, the nature of FAM83-CK1 α might preclude CK1 α from being recognised by CRBN upon IMiD treatment. We discuss these possibilities in the Discussion section. We attempted to analyse endogenous CRBN IPs for interaction with CK1 α , FAM83F, ZFP91 and FAM83G in the presence of IMiDs but were unable to detect any proteins, not even the positive controls (CK1 α , FAM83F, and ZFP91). Nonetheless, a study in which interactors of overexpressed CRBN upon lenalidomide treatment were analysed by mass spectrometry, identified FAM83G and CK1 α among the most abundant proteins, suggesting that FAM83G-CK1 α complex is likely to be recruited by CRBN upon IMiD treatment [2]. Excitingly, this raises the possibility that unique variants of IMiDs, that modulate the positioning of the CRBN-recruited FAM83-CK1 α complexes, might be able to target other FAM83-CK1 α complexes resulting in ubiquitylation and degradation of either FAM83 or CK1 α , or even both. For example, in the case of the FAM83G-CK1 α complex, knocking down either component results in the destabilisation of the other [3].

Response Figure 1: Pomalidomide treatment does not affect the interaction between FAM83G and CK1α. Lysates from HCT116 wild-type and FAM83G^{-/-} cells treated with 10 μM pomalidomide for 24 h, were subjected to immunoprecipitation with anti-FAM83G antibody. Input lysates and FAM83G IP elutes were resolved on SDS-PAGE and subjected to western blotting with indicated antibodies.

2) *In my view, one of the most striking observations of this study is the clear difference between the poor degradation of CK1α in most cell lines (except MV4.11), in comparison to the very strong degradation of FAM83F. These results should be more clearly highlighted and discussed.*

Response: We agree with the reviewer and we have endeavoured to highlight this point throughout. Nonetheless, we feel it is important to explain this absence of robust IMiD-induced CK1α degradation in many cell lines given what we know about the existence of eight FAM83-CK1α complexes that define a context for CK1α in different cells, and we highlight this in our manuscript as well.

We have expanded our discussion section to highlight the observed differences in CK1α degradation between cell lines with the addition of the following text: “We propose that the efficiency of CK1α degradation is rather influenced by the relative abundance of FAM83 proteins. We observe substantial lenalidomide-induced CK1α degradation in MV4.11 cells, which lack expression of several FAM83 proteins, but not in any other cell line which display higher abundance of FAM83 proteins. Therefore, we hypothesise that lenalidomide can facilitate the degradation of the non-FAM83 bound pool of CK1α as well as the FAM83F-bound pool and thus the abundance of other FAM83 proteins in cells may be used as predicting biomarkers for levels of IMiD-induced CK1α degradation, which may inform the use of lenalidomide for targeting CK1α.”

3) *One serious issue: Stats are absent from most blot quantifications, including those from Fig.1 on which the whole work is based. Without proper statistical analysis, these quantifications have little value.*

Response: We thank the reviewer for bringing this to our attention. We have added statistical analyses to all quantification plots.

4) *Along the same lines, in those panels where stats are included, one sees p values of 0.08 and 0.058, as well as several cases with a single * ($p < 0.05$). Although it is completely fine to show such values, it is very dangerous to extract conclusions out of such data. There is very good literature explaining how "weak" is the significance of a $p = 0.05$.*

Response: We thank the reviewer for highlighting this important point. We have adjusted the text regarding descriptions of borderline significant p-values to highlight whether the data is statistically significant. Specifically, we have added following text when discussing Figures 5E&F which contains a p-value of 0.05: "However, it must be noted that these changes in protein abundance are only slight and not statistically significant, which is unsurprising as CK1 α exists in multiple protein complexes, including in other FAM83-CK1 α complexes."

5) *The choice of DLD-1 cells, which have a mutated APC, as model to test the impact on Wnt pathway is interesting for potential therapeutic applications, but biologically, the rationale is unclear. One would wish a) a detailed rationale which will take into account the known composition/function of the Axin-APC complex and of the position of CK1a in this context. b) Verification of the effect on Wnt signalling in a cell line with a functional pathway.*

Response: The reasons for choosing DLD-1 cells were two-fold: Firstly, they expressed detectable levels of FAM83F protein which was not apparent among other cells we scanned. Secondly, as discussed by the reviewer, DLD-1 cells contain an Apc mutation and display hyperactive Wnt signalling. FAM83F modulates Wnt signalling. Given the prevalence of Apc mutations in colorectal cancer (CRC), investigating the impact of FAM83F-CK1 α degradation by IMiDs on Wnt signalling in DLD-1 cells could potentially inform whether IMiDs could reduce Wnt signalling in cells with Apc mutant backgrounds. We have added the following text to explain our rationale on page 9: "Hyperactivated Wnt signalling caused by a truncation of adenomatous polyposis coli (Apc) is a hallmark of colorectal cancer initiation and thus we sought to determine if IMiD-induced FAM83F degradation could dampen down Wnt signalling in DLD-1 cells, which harbour a truncated Apc mutant protein [4]."

The reviewer raises an important question regarding Wnt signalling biology which is ultimately affected in these Apc mutant cells. Regarding the Axin-Apc complex in DLD-1 cells, it has been demonstrated that truncation of Apc results in the loss of multiple β -catenin and Axin binding sequences but interestingly truncated Apc can still interact with components of the β -catenin destruction complex; specifically Axin, β -catenin and GSK-3 β [5]. Whilst the interaction with CK1 α was not assessed in this publication, the phosphorylation of β -catenin at serine 45 by CK1 α is still observed in DLD-1 cells containing mutant Apc [4], indicating that the β -catenin destruction complex is still functional, to some extent, in DLD-1 cells. It has recently been shown that Apc truncations can prevent the recruitment of the β -catenin destruction complex to the local Wnt3A signal in CRC cells [6]. However, whilst recruitment of Apc, Axin and GSK-3 β to the Wnt3A signal was impaired in

DLD-1 cells, CK1 α was still able to localise to this local Wnt3A signal albeit at lower levels compared to cells containing wild-type Apc [6]. This suggests that the localisation of CK1 α in Wnt signalling may be in part regulated by factors other than the β -catenin destruction complex, which we hypothesise to be the FAM83 proteins [7].

The detailed role of FAM83F-CK1 α in mediating Wnt signalling is described in a separate manuscript, which was co-submitted to Life Science Alliance and should have been made available to the reviewer [7]. Nonetheless, for clarity we have added the following in the Discussion section: "Whilst there are benefits to testing the ability of IMiD-induced FAM83F degradation to attenuate Wnt signalling in cells containing Wnt activating mutations, the presence of truncated Apc adds increased complexity. Apc is a key component of the β -catenin destruction complex which regulates the canonical Wnt signalling pathway by regulating levels of the effector protein, β -catenin, and is required for Wnt signalosome formation following Wnt ligand binding [6, 8]. The presence of an Apc truncation mutation disrupts this complex to such an extent as to increase Wnt signalling but not completely abolish the complex, so that in DLD-1 cells, truncated Apc can still interact with other components of the β -catenin destruction complex and phosphorylation of β -catenin (S45) by CK1 α is still present [4, 5]. Therefore, there are caveats to be considered when evaluating Wnt signalling effects induced by removal of FAM83F-CK1 α complexes in these cell lines."

To test if pomalidomide could reduce Wnt signalling in cells which have no known mutations within the Wnt signalling pathway, we treated HaCaT, HEK-293 and A549 cells with pomalidomide and measured *Axin2* mRNA expression and FAM83F protein degradation (**Figure 7C&D**). Pomalidomide treatment reduces FAM83F protein levels and significantly reduces basal *Axin2* transcript abundance in HaCaT cells with no changes detected following Wnt3A-CM exposure. Pomalidomide treatment increased *Axin2* transcripts in HEK-293 cells treated with L-CM but not Wnt3A-CM. This is likely a consequence of the modest CK1 α degradation observed in HEK-293 cells (**Figure 7D**). There was no reduction in *Axin2* transcripts observed in A549 cells after exposure to L-CM or Wnt3A-CM. Interestingly, neither A549 cells nor HEK-293 cells contain detectable levels of FAM83F protein suggesting that the reduction of *Axin2* by pomalidomide appears to require FAM83F protein in cells.

6) *Figure 2: The IF signals are very weak, in particular for CK1 α . Please increase contrast and perhaps show the CK1 α channel in gray scale for better visualization.*

Response: We have adjusted the contrast of both the GFP and CK1 α channels to improve visualisation.

- [1] Donovan KA, An J, Nowak RP, Yuan JC, Fink EC, Berry BC, Ebert BL, Fischer ES. Thalidomide promotes degradation of SALL4, a transcription factor implicated in Duane Radial Ray syndrome. *Elife* 2018;7.
- [2] Kronke J, Udeshi ND, Narla A, Grauman P, Hurst SN, McConkey M, Svinkina T, Heckl D, Comer E, Li X, Ciarlo C, Hartman E, Munshi N, Schenone M, Schreiber SL, Carr SA, Ebert BL. Lenalidomide causes selective degradation of IKZF1 and IKZF3 in multiple myeloma cells. *Science* 2014;343(6168):301-5.
- [3] Bozatzki P, Dingwell KS, Wu KZ, Cooper F, Cummins TD, Hutchinson LD, Vogt J, Wood NT, Macartney TJ, Varghese J, Gourlay R, Campbell DG, Smith JC, Sapkota GP. PAWS1 controls Wnt signalling through association with casein kinase 1alpha. *EMBO Rep* 2018;19(4).
- [4] Yang J, Zhang W, Evans PM, Chen X, He X, Liu C. Adenomatous polyposis coli (APC) differentially regulates beta-catenin phosphorylation and ubiquitination in colon cancer cells. *J Biol Chem* 2006;281(26):17751-7.
- [5] Li VS, Ng SS, Boersema PJ, Low TY, Karthaus WR, Gerlach JP, Mohammed S, Heck AJ, Maurice MM, Mahmoudi T, Clevers H. Wnt signaling through inhibition of β -catenin degradation in an intact Axin1 complex. *Cell* 2012;149(6):1245-56.
- [6] Parker TW, Neufeld KL. APC controls Wnt-induced β -catenin destruction complex recruitment in human colonocytes. *Sci Rep* 2020;10(1):2957.
- [7] Dunbar K, Jones RA, Dingwell K, Macartney TJ, Smith JC, Sapkota GP. FAM83F regulates canonical Wnt signalling through an interaction with CK1 α . 2020 (Unpublished).
- [8] Clevers H, Nusse R. Wnt/ β -catenin signaling and disease. *Cell* 2012;149(6):1192-205.

December 8, 2020

RE: Life Science Alliance Manuscript #LSA-2020-00804-TR

Dr. Gopal P. Sapkota
University of Dundee
MRC Protein Phosphorylation and Ubiquitylation Unit
School of Life Sciences
Dow Street
Dundee, Scotland DD1 5EH
United Kingdom

Dear Dr. Sapkota,

Thank you for submitting your revised manuscript entitled "IMiDs induce FAM83F degradation via an interaction with CK1 α to attenuate Wnt signalling". We would be happy to publish your paper in Life Science Alliance pending final revisions necessary to meet our formatting guidelines.

Along with the points listed below, please also attend to the following,

- please add callouts for Figure 8A, 8B, and 8C
- please reformat the reference style to 10 authors et al

A. FINAL FILES:

-- Summary blurb (enter in submission system): A short text summarizing in a single sentence the study (max. 200 characters including spaces). This text is used in conjunction with the titles of papers, hence should be informative and complementary to the title. It should describe the context and significance of the findings for a general readership; it should be written in the present tense

and refer to the work in the third person. Author names should not be mentioned.

B. MANUSCRIPT ORGANIZATION AND FORMATTING:

Sincerely,

Shachi Bhatt, Ph.D.
Executive Editor
Life Science Alliance
<https://www.lsjournal.org/>
Tweet @SciBhatt @LSAJournal

Reviewer #3 (Comments to the Authors (Required)):

The authors have satisfactorily answered all queries. The revised manuscript presents a high quality study of an interesting regulatory mechanism.

December 11, 2020

RE: Life Science Alliance Manuscript #LSA-2020-00804-TRR

Dr. Gopal P. Sapkota
University of Dundee
MRC Protein Phosphorylation and Ubiquitylation Unit
School of Life Sciences
Dow Street
Dundee, Scotland DD1 5EH
United Kingdom

Dear Dr. Sapkota,

Thank you for submitting your Research Article entitled "IMiDs induce FAM83F degradation via an interaction with CK1 α to attenuate Wnt signalling". It is a pleasure to let you know that your manuscript is now accepted for publication in Life Science Alliance. Congratulations on this interesting work.

*****IMPORTANT:** If you will be unreachable at any time, please provide us with the email address of an alternate author. Failure to respond to routine queries may lead to unavoidable delays in publication.*******

DISTRIBUTION OF MATERIALS:

Again, congratulations on a very nice paper. I hope you found the review process to be constructive and are pleased with how the manuscript was handled editorially. We look forward to future exciting

submissions from your lab.

Sincerely,

Shachi Bhatt, Ph.D.

Executive Editor

Life Science Alliance

<https://www.lsjournal.org/>
